# Embryos assist morphogenesis of others through calcium and ATP signaling mechanisms in collective teratogen resistance

Angela Tung[1], Megan M. Sperry[1,2], Wesley Clawson[1], Ananya Pavuluri[1], Sydney Bulatao[1], Michelle Yue[2], Ramses Martinez Flores[2], Vaibhav P. Pai[1], Patrick McMillen[1], Franz Kuchling[1] & Michael Levin[1,2] ✉

Information for organismal patterning can come from a variety of sources. We investigate the possibility that instructive influences for normal embryonic development are provided not only at the level of cells within the embryo, but also via interactions between embryos. To explore this, we challenge groups of embryos with disruptors of normal development while varying group size. Here, we show that *Xenopus laevis* embryos are much more sensitive to a diverse set of chemical and molecular-biological perturbations when allowed to develop alone or in small groups, than in large groups. Keeping per-embryo exposure constant, we find that increasing the number of exposed embryos in a cohort increases the rate of survival while incidence of defects decreases. This inter-embryo assistance effect is mediated by short-range diffusible signals and involves the P2 ATP receptor. Our data and computational model emphasize that morphogenesis is a collective phenomenon not only at the level of cells, but also of whole bodies, and that cohort size is a crucial variable in studies of ecotoxicology, teratogenesis, and developmental plasticity.

During embryogenesis, a complex anatomical form is constructed via the interactions of large numbers of individual cells. A key question, with impacts ranging across evolution, regenerative medicine, and bioengineering, concerns the origin of the information enabling correct morphogenesis. Most of the emphasis to date has been placed upon the information provided vertically from parent to offspring–the genome–and the inanimate environment[1–8]. The typical perspective is of a single embryo and its local microenvironment as the arena within which cells compete and cooperate using their genomically specified cellular hardware to complete embryogenesis in a given situation. While work on developmental plasticity and extended phenotype do consider factors outside of individual bodies[9–13], the embryo (and its outer perimeter) is most

commonly taken to be the natural, self-contained unit of studies on control mechanisms and the origin of specific anatomies in evolutionary morphology, reproductive toxicology, and developmental genetics. However, many phenomena in biology exhibit scale-free or at least multi-scale dynamics[14–16]. Thus, we explored the possibility of relationships between development and external social environments, specifically whether instructive information could also propagate horizontally, enabling embryos to benefit from a kind of "wisdom of the crowd" in their cohort[17–20].

While the focus of most studies has been the self-contained genome, a number of prior studies have also reported extra-genomic influences on developmental outcomes[21], in the context of phenotypic plasticity[11,22–24]. Of particular interest were studies focusing on

[1]Allen Discovery Center at Tufts University, Medford, MA 02155, USA. [2]Wyss Institute for Biologically Inspired Engineering, Harvard University, Boston, MA 02115, USA. ✉e-mail: michael.levin@tufts.edu

inter-conspecific interactions or interactions amongst members of the same cohort. For example, rat studies have shown that the uterine environment and position in the uterine horn can affect progeny. Females that were located contiguous to a caudal male or between two males exhibited more masculine characteristics, mounting behavior, and morphology than females contiguous to a cephalic male or near no males[25,26]. Adverse intrauterine conditions lead to females being bigger than their male littermates and increased uterine occupancy leads to smaller males[27]. In humans, studies have shown that skin-to-skin contact between caregivers and preterm infants can have positive cardio-respiratory impacts[28,29]. Inter-embryo communication in the context of predation has been observed in a number of species ranging from birds to frogs[30–34].

These lateral interactions go beyond straightforward negative effects such as competition for nutrients, oxygen levels, and accumulating waste products, and suggest that there are also beneficial effects from being part of a collective. Postnatally, the Allee effect[35] is a well-known effect of animal aggregation and its correlation with fitness. It is defined as a positive correlation between population density and individual fitness. Studies in fish, rodents, and planaria have shown that having conspecifics can positively impact future health outcomes. For example, large groups of planaria and goldfish survive colloidal silver exposure better than small groups. In line with other studies showing the beneficial effects of larger groups, dense populations of starfish were able to right themselves faster than sparse populations[35–38].

These effects are seen at the cellular level as well, from social amoebas[39] to metazoan cells in culture[40]. In mesenchymal cells in vitro, low population seeding impacts the differentiation potentials and alters cell fates and growth rates of tumors, all of which are affected by cell density[41,42]. It has long been realized that groups of cells resist transformation and cancer[43–46], providing a collective dynamic that keeps cell activity orchestrated towards adaptive outcomes[47]. One such example of cellular resistance is the culturing of multiple ovarian follicles, which improves follicular survival and growth through paracrine signaling from one follicle to another[48]. In the case of tissue injury, paracrine factors have also been shown to play an important role. In *Xenopus* embryos, wounding induces calcium waves within a single organism that aid in cell-cell coordination and permit rapid wound closure in response to superficial injury[49,50]. ATP is one such ligand that can trigger cell contractility and plays a role in coordinating long-range contraction responses after injury[51]. However, the transfer of such information between individuals or distinct tissues and organs has not been visualized or quantified.

Thus, a number of cases of beneficial lateral interactions have been reported, and the study of cell cooperativity and instructive communication is an active field. However, a significant knowledge gap exists with respect to *instructive lateral interactions* for correct development at a level of organization above that of tissues, organs, and individual organisms. While epigenetic studies have explored how genes can be impacted on environment and behavior, the information transfer from this field is still vertical. Here, we sought to test the hypothesis that embryos can interact across distance in their medium to provide beneficial, specific information that assists morphogenesis. Our fundamental assay was resistance to teratogens: we exposed groups of *Xenopus laevis* embryos to several disruptors of normal development with diverse mechanisms of action and asked whether large groups can resist exposure better than small ones. We found that collective development is much more stable, with respect to morphological perturbations (birth defects), and analyzed an agent-based cellular automata computational model that explains the collective morphogenetic stability of larger embryonic groups through local interactions. Functional experiments reveal that this effect is mechanistically mediated by a short-range chemical signal, requiring calcium and P2 receptor signaling. It also does not involve detectable

transcriptional changes at stage 25, but does implicate a small group of changes by stage 35. We also report the remarkable phenomenon of a mechanical injury-triggered calcium wave that propagates not only within embryos, as has been shown previously[49–51], but also between embryos, identifying it as a potential candidate for the communication process within a network of developing individuals. Finally, our transcriptomic profiling of small vs. large groups identifies a set of up- and down-regulated genes that provide a signature of the collective response to teratogen challenge.

## Results

### Incidence and severity of thioridazine-induced defects are dependent on group size

To test the hypothesis of collective influences on development, we asked whether group size was a factor in embryos' ability to undergo normal morphogenesis despite the presence of well-known teratogenic influences. Embryos for each experiment were from the same clutch but separated into differing group sizes. Importantly, we kept teratogen concentration constant between groups by scaling media volume to group size so that the per-embryo teratogen exposure was constant regardless of group size.

We began with thioridazine, a chemical reagent that targets dopamine pathways[52,53]. *Xenopus laevis* were raised in identical, standard conditions but at different group sizes within each Petri dish, and subjected to 90 μM of thioridazine across all densities (Fig. 1a). Media and dish size were scaled proportionally between large and small groups so that the only difference between experimental conditions was the number of animals in a dish. Large groups ($n = 300$) received 120 mL of media while small groups ($n = 100$) received 40 mL of media. To rule out greater drug degradation by larger cohorts as a potential confounder, we quantified thioridazine in the medium following the treatment of embryos in groups of various sizes. Liquid chromatography–mass spectrometry analysis indicated that there was no notable difference in thioridazine concentration between groups ranging in size from 25 to 400 (Fig. 1b) beyond that attributable to compound degradation during the preparation and transport of samples.

Following treatment with 90 μM thioridazine for 18 h or stages 12.5–25, we observed both embryonic death and a range of defects in survivors and scored their incidence. Survival increased with increasing group size (Fig. 1c). For singlets and groups of 5 embryos, thioridazine exposure resulted in 0% survival. When exposed in groups of 25 and 75, averages of 13% and 44% of embryos survived, respectively. For group sizes of 100 and 300, survival rates increased to averages of 85% and 98%, respectively ($N = 3$, $p = 0.0051$ using ANOVA). Among survivors, thioridazine caused significant developmental abnormalities, as expected. Compared to age-matched controls, exposure to thioridazine resulted in a rectangular head shape, misshapen eyes, and hyperpigmentation (Fig. 1d and Supplementary Fig. 1A, B). After exposure, tadpoles in the largest group treatment ($n = 300$) and smaller group treatment ($n = 100$) were scored at NF stage 45 for defects (Fig. 1e). Large groups had significantly fewer average incidents of square head (22.7% vs. 40.7%) (see Methods section for quantitative procedure for assessing phenotype) and hyperpigmentation (37.8% vs. 54.2%) ($N = 6$, $p = 0.00029$ using Welch's $t$-test). While other types of phenotypic defects were observed (Supplementary Fig. 1C), here we chose to focus on head shape and pigmentation for the in-depth analysis of cross-embryo effects.

As quantitative validation that individuals scored with the hyperpigmentation phenotype were truly different from controls, 10 tadpoles were pooled, dissolved, and measured for absorbance at 260 nm. Control animals had an average absorbance of 0.427 while hyperpigmented animals had an average absorbance of 0.615 (Fig. 1f) ($N = 3$, $p = 0.0029$ using Welch's $t$-test). We used morphometric analysis to further quantify the square head shape phenotypes by

determining the difference between the diameter of the head and the width at the base of the branchial arch (Fig. 1g). In treated animals with square heads, this number is significantly smaller than in controls (261.2 vs. 23.5, respectively. $N = 3$, $p < 0.0001$ using Welch's $t$-test)

(Fig. 1h). To prevent bias, blind scoring was used for three of the six experimental replicates of thioridazine.

We conclude that larger group size confers a significant protective effect against both thioridazine-induced death and craniofacial defects

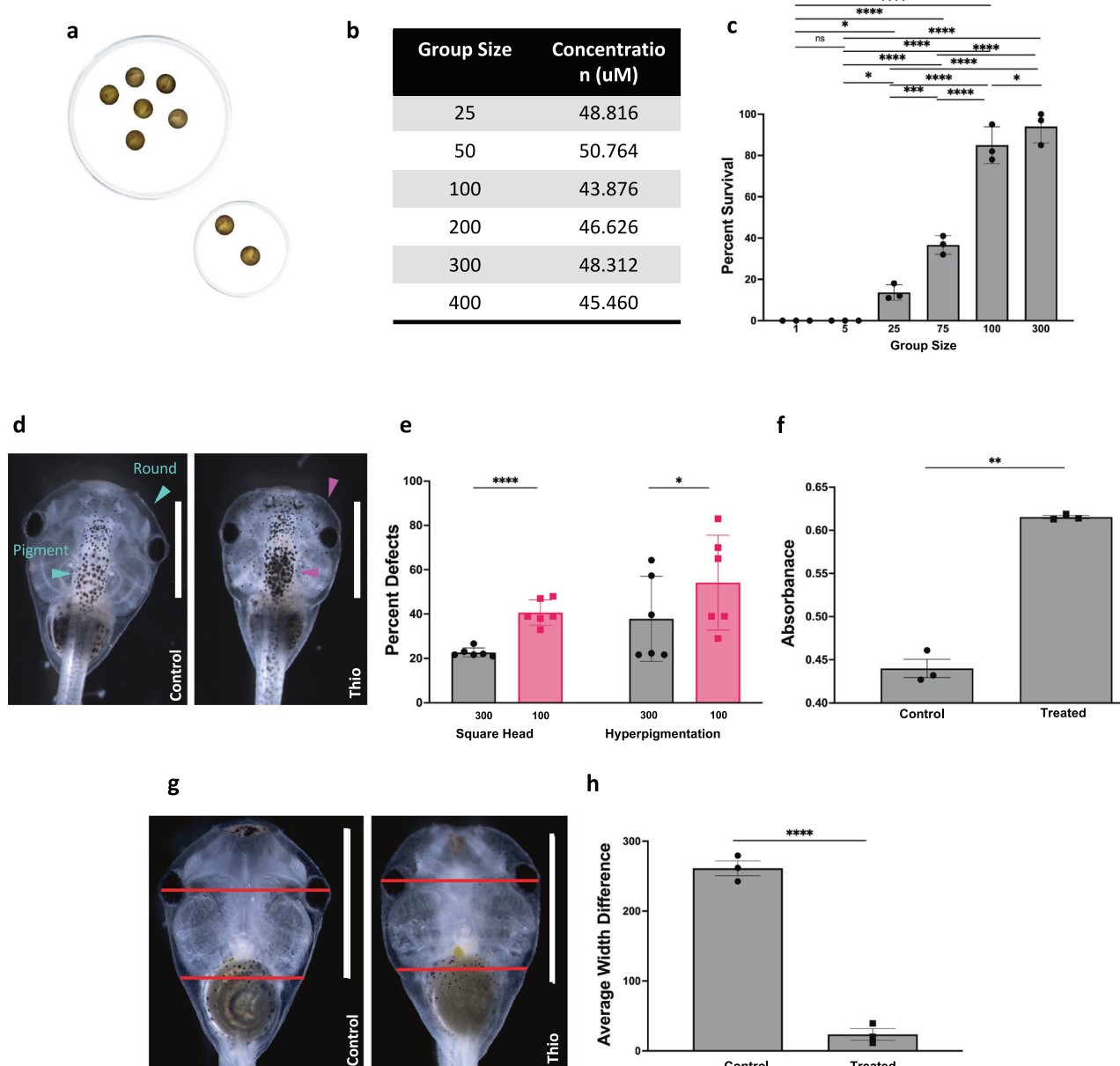

**Fig. 1 | Conspecifics help resist teratogenic effects. a** Embryos were reared in large ($n = 300$) or small ($n = 100$) groups and exposed to the dopaminergic agent thioridazine during N&F stages (13-26). The concentration of the drug was kept equal at 0.4 µg/mL of media and the amount of media with the drug was scaled according to the group size to match the volume of the drug per embryo. **b** Embryos were treated with thioridazine in various group sizes. Post-treatment, media were harvested and thioridazine concentration was determined by LC-MS. The table shows the concentration of thioridazine in media from embryos treated in groups of 25, 50, 100, 200, 300, and 400. **c** Percentage of surviving animals in groups of increasing size from 1 to 300 following a 24-h treatment with 90 µM thioridazine. Data was compared using a one-way ANOVA test and Tukey test; points are the average of 3 replicates. ****$p < 0.0001$, **$p = 0.0024$. **d** Craniofacial phenotype of a normal tadpole and a thioridazine-treated tadpole at stage 45. Control embryo with normal, round head shape and normal pigmentation (blue arrows). Thioridazine-treated embryo with square head shape and hyperpigmentation (purple arrows). **e** Total frequency of malformed square heads, and

hyperpigmentation in large group thioridazine exposure ($n = 300$ in gray) and small group treatment ($n = 100$ in pink) across $N = 6$ trials. Comparisons were done using two-tailed Welch's $t$-test with $p = 0.0003$ in square heads and $p = 0.0489$ in hyperpigmentation. **f** Quantification of hyperpigmentation between untreated (control) and thioridazine-treated hyperpigmented animals. A total of 30 control and 30 thioridazine-treated embryos were used in each trial. Two-tailed Welch's $t$-test was used for comparison and $p = 0.0029$. **g** Ventral view of normal and square-headed animals with lines indicating where width measurements were taken. **h** Quantification of face width difference between untreated (left) and treated square head (right) animals. A total of 30 square head and 30 control embryos were examined for each trial. Comparison is done with a two-tailed Welch's $t$-test with a $p < 0.001$. For all animal images, the anterior end is at the top of the image. All images unless noted are dorsal views. Data are plotted as mean ± SD and each point represents a trial. All treatments are normalized to the mean of non-treated controls' spontaneous defects. Scale bars represent 2 mm. Source data are provided as a Source Data file.

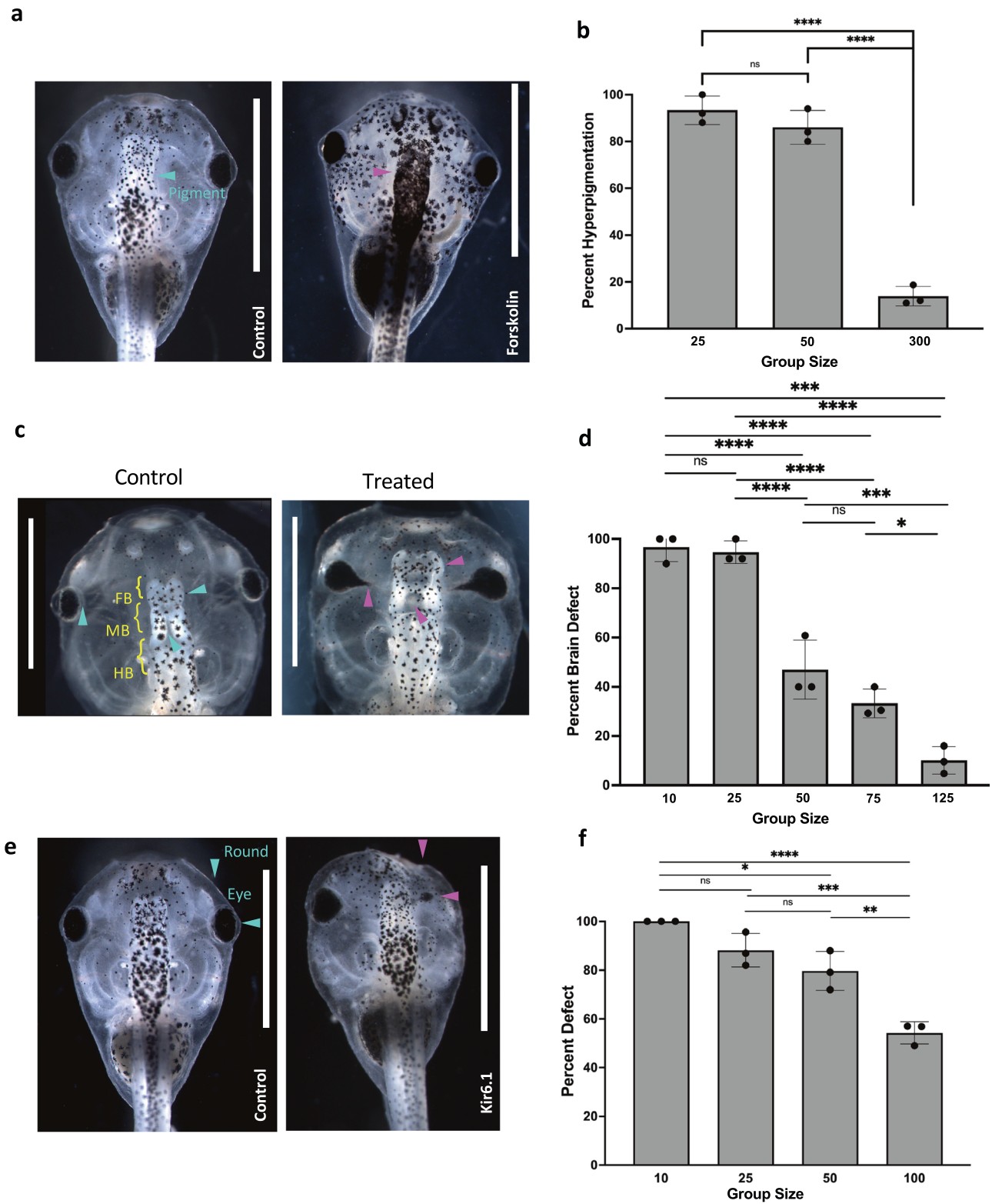

in a manner that cannot be explained by increased drug degradation in larger cohorts. We call this phenomenon the Cross-Embryo Morphogenetic Assistance (CEMA) effect.

## Cross-embryo morphogenetic assistance effect is not limited to thioridazine exposure

We next asked whether the surprising CEMA effect is a feature of thioridazine teratogenesis specifically or whether it is a wider phenomenon, and, if so, what kinds of perturbations can embryos resist collectively better than singly? We explored additional teratogens orthologous to thioridazine: other drugs with a different target, and a non-pharmacological method using misexpression of mutant mRNA.

Two drugs with different targets from each other and from thioridazine were used to test the robustness of the CEMA effect. We first tested forskolin—a drug that targets adenylate cyclase and results

**Fig. 2 | Conspecific effect beyond thioridazine exposure. a** Images of control (left) and forskolin-treated animals (right), demonstrating hyperpigmentation following forskolin exposure (purple arrows). **b** The graph shows the total frequency of hyperpigmentation in embryos exposed to forskolin in small (*n* = 25), medium (*n* = 50), and large (*n* = 300) groups. ****p < 0.001.. **c** Control animal raised in 0.1x MMR with normal brain structures (left). ****p <0.0001. Representative animals from cohorts subjected to nicotine treatment were then raised at a group size of 75 animals (right). Purple arrows highlight misshapen eyes, malformed forebrain, and midbrain. **d** Percent total of embryos with brain defects in groups treated with nicotine at densities of n = 10, 25, 50, 75, and 125. ****p < 0.0001, ***p = 0.0008, *p = 0.0197. **e** Effect of group size on incidence of defects in dominant-negative Kir6.1-injected embryos. Images of control (left) and injected (right) embryos at stage 45. Blue arrows indicate normal eyes and head shape while purple arrows point out misshapen heads and eyes. **f** Quantification of malformed and normal individuals at stage 45. Group size of *n* = 10, 25, 50, and 100. Both cells at the 2-cell stage were injected and embryos were split into different size groups post injection. *p = 0.0111, ****p < 0.0001, ***p = 0.0004, and **p = 0.0028. All animal images are dorsal views and oriented so anterior is facing up and dorsal is down. For all data: values are plotted as mean ± SD, one-way ANOVA, and Tukey tests were conducted. Each dot on the graph is a separate replicate. All treatments are normalized to the mean of non-treated controls' spontaneous defects. Scale bars represent 2 mm. Source data are provided as a Source Data file.

in hyperpigmentation (Fig. 2a and Supplementary Fig. 2B)[54]. Tadpoles were subjected to 5 µM of forskolin at stages 10–45 in group sizes of small (*n* = 25), medium (*n* = 50), and large (*n* = 300) (Fig. 2b). In the small and medium groups, forskolin resulted in hyperpigmentation in an average of 93.3% and 86% of embryos, respectively, while the large group exhibited hyperpigmentation in an average of just 13.9% of embryos (*N* = 3, *p* < 0.001 using ANOVA). Blind scoring was done in two of three replicates to prevent any bias.

Next, we tested nicotine, which is known to cause developmental brain and other defects[55–57]. Embryos were separated into different group sizes at stage 11 and treated with nicotine until stage 35, and then allowed to develop in the regular medium until stage 45. Animals were then scored for abnormalities including missing or malformed eyes, fusion of eye to the brain, and brain defects including absence of forebrain, missing both forebrain and midbrain, malformed hindbrains, or incorrect separation of the forebrain and midbrain (Fig. 2c and Supplementary Fig. 2C). For tadpoles reared and exposed in the smaller groups (*n* = 10 and *n* = 25), an average of 96.7% and 94.7%, respectively, exhibited brain defects. Those reared at higher densities (50 and 75) had defects in an average of 46.9% and 33.3% of embryos, respectively. At the highest density, *n* = 125, only an average of 10.1% of embryos exhibited defects (*N* = 3, *p* < 0.001, ANOVA) (Fig. 2d).

We next examined the effect of group size on a mechanistically different kind of perturbation that has previously been shown to cause defects in eye and heart morphogenesis[56]: disrupting native bioelectric signaling among embryonic cells[58–60] by microinjecting mRNA encoding a mutant ion channel. We found that large group size also protected embryos against the negative effects of mutant ion channel expression. At the 2-cell stage, both cells were microinjected with mRNA encoding dominant-negative Kir6.1[61–63]. Following injection, animals were raised in dishes containing varying embryo group sizes. Defects observed at stage 45 included misshapen eyes, overall head malformations, hyperpigmentation, and death (Fig. 2e and Supplementary Fig. 2D). Unlike thioridazine, dominant-negative Kir6.1 microinjections did not induce significant embryo death and we saw no significant differences in survival between embryos raised in different group sizes and controls. There was, however, again a dramatic protective effect of group size on the incidence of craniofacial defects. At a density of 5 embryos per dish, 100% of the injected embryos had defects. Increasing the density to 25 and 50 animals lowered the average defect incidence to 88.2% and 79.7%, respectively. At the highest group size, *n* = 100 animals, the average defect incidence was 54.2% and the average of normal individuals was 43.0% (*N* = 3, *p* = 0.0141 using ANOVA) (Fig. 2f). We note that the mRNA microinjection experiments rule out explanations based on the pharmacokinetics of drugs in different sized groups. Because there is no significant embryo death, they also indicate that the protective effect of larger group size against craniofacial defects is not secondary to an effect on survival.

These results indicate that the protective effect of larger group size on craniofacial defects is not specific to one kind of pathway or teratogen. We conclude that CEMA is a more general developmental

stability mechanism that can stabilize several diverse kinds of developmental processes against perturbations.

## CEMA operates across genetically diverse populations, but only among perturbed individuals

We next asked whether the assistive effect is only operational across identical conspecifics, or whether genetically diverse groups also reap the benefits. We mixed strains of wild-type and an albino strain of Xenopus laevis, then treated them to ask whether one genetic background can help another (Fig. 3a). A group of 150 wildtype embryos treated with 90 µM thioridazine (stages 12.5–25) had a square head average of 44.7% and 44.3% for 150 treated albino embryos. Increasing the group size to 300 leads to averages of 20.7% for wild types and 23.7% for albinos. Mixing a group of 150 wildtypes and 150 albino embryos had an average of 21% (Fig. 3b). While there were significant differences between the 150 and 300 group sizes, there were no significant differences between wildtype and albino embryos of the same group size. Given this, we conclude that CEMA does not require the large cohort to be genetically homogenous.

In prior experiments, every animal in a given cohort was exposed to the same stressor. We next asked whether animals that had never experienced the teratogen could improve the resistance of conspecifics that were exposed. This would be expected if, for example, animals were providing developmental signals to each other–in that case, unperturbed control embryos would be even more efficient than exposed ones in stabilizing other embryos. Thus, we investigated CEMA in experiments in which wild-type animals exposed to thioridazine were reared with naïve, untreated albino embryos in mixed cohorts. After treatment with thioridazine, wild-type animals were rinsed three times before being mixed in with an equal number of untreated albinos. Consistent with our previous experiments, groups of 150 treated wildtype animals had an average square head incidence of 36.0%, which was significantly reduced to an average of 23.3% in groups of 300 treated wildtype animals (*N* = 3, *p* = 0.0068, ANOVA). However, in the mixed cohort (150 treated wildtype animals with 150 untreated albinos), an average of 37.0% had defects, not significantly different from that seen in the group of 150 treated embryos reared alone (Fig. 3c). The 300 mixed group had no significant difference from that of the 150 treated wildtype group, but there was a significantly increased incidence of square heads compared to that of the 300 all treated group (*N* = 3, *p* = 0.0068 using ANOVA). Thus, unperturbed individuals do not help stabilize teratogen-exposed individuals and we conclude that only perturbed individuals have a role in CEMA.

In order to distinguish whether CEMA contributes generic protective influence or encodes morphogenetic information for a specific defect, a cross-teratogen experiment was performed using nicotine and thioridazine. Nicotine primarily induces brain defects while thioridazine induces head shape, eyes, and pigmentation defects. Embryos were treated with either nicotine or thioridazine and then mixed together. If CEMA was a generic effect, then it could be expected that groups made up of cohorts of embryos exposed to different insults would function as a large group–exhibit CEMA-

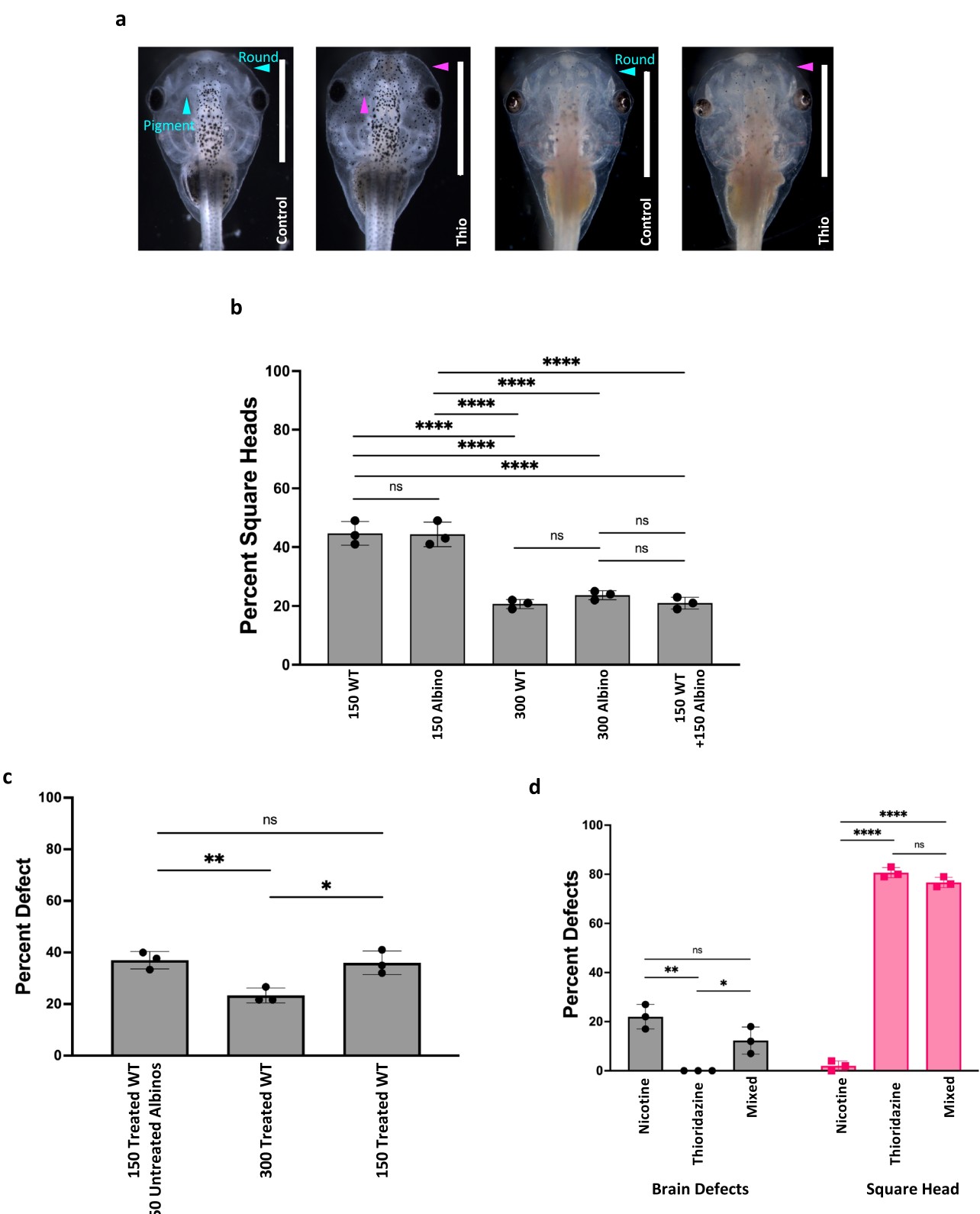

induced mitigation of the phenotypes. In contrast, if each type of insult involved phenotype-specific signals spread through the population, then such chimeric groups would not be additive in the way that CEMA requires, because each specific insult cohort would be too small to exhibit significant CEMA-induced repair. We observed that in a group of 100 animals, there was an average of 22% in normal nicotine treatment, 0% in normal thioridazine treatment, and 12.3% in nicotine

treatment followed by mixing in with thioridazine conspecifics. With regards to square heads, nicotine treatment alone resulted in 2%, thioridazine alone resulted in 80.7%, and mix groups resulted in 76.7% (Fig. 3d). While there are decreases in the mixed group compared to either the normal nicotine or thioridazine treatment, the decreases are non-significant, suggesting that CEMA does not operate when the large group is made up of subgroups with a different history of teratogen

**Fig. 3 | Genetically diverse populations can also benefit from CEMA but unperturbed cohort members do not aid in stabilization. a** Examples of wildtype and albino animals that were untreated (controls) or treated with thioridazine (treated). Blue arrows highlight normal head shape and pigmentation. Purple arrows indicate square heads and hyperpigmentation. **b** Percent total of embryos with square heads in two strains of Xenopus laevis, wildtype (either $n = 150$ or $n = 300$, respectively labeled on the graph) and albino ($n = 150$ or $n = 300$, respectively labeled on the graph). ****$p < 0.0001$. **c** Averages of the frequency of square head defects in a mixed group of $n = 150$ thioridazine-treated wildtype embryos +

150 untreated albinos, $n = 300$ treated wild-type embryos, and $n = 150$ treated wild-type embryos. **$p = 0.0094$ and *$p = 0.0133$. **d** Average of the frequency of brain defects and square heads in nicotine, thioridazine, and mixed treatments ($n = 150$ in each group). **$p = 0.0019$ and *$p = 0.0291$ for brain defects and ****$p < 0.0001$ for square head. All animal images are dorsal views with the anterior end at the top of the image. For all data: values are plotted as mean ± SD, one-way ANOVA, and Tukey test were conducted. All treatments are normalized to the mean of non-treated controls' spontaneous defects. Scale bars represent 2 mm. Source data are provided as a Source Data file.

exposure. These data are consistent with individual stressors eliciting their own specific repair information which is not additive and does not cross-protect.

## A computational model of CEMA

To better understand inter-embryonic signaling within CEMA and guide future studies, we developed a model of the system-level collective dynamics and ran simulations to probe potential explanations and generate testable predictions. Specifically, we developed a computational approach to show how the development of embryos (or any morphogenic agent) may be more stable and harder to disrupt in larger groups than for singletons or small groups. We chose an agent-based approach to virtual embryogeny[64–69], where each embryo was a cellular automaton[70–77] that could interact with its neighbors through diffuse signaling and was parametrized to reflect the quantitative data in Fig. 1 with regard to survival as a function of cohort size.

For our model, we chose to use elementary cellular automata (ECA), which are 1D arrays consisting of cells that can hold a value of either 0 or 1[78]. At each timestep in a simulation, these cells can change their values based on pre-programmed update rules, wherein they sample their own value and their close neighbor's (within 1 to 3 cells to the left or right) and either maintain their existing value (0 or 1) or change to the opposite value. This inherent 'local interactions only' property makes ECAs a good choice for this model since many biological collective systems (especially during morphogenesis) appear to be governed by parallel local interactions not dependent on a central controller[79–81].

We chose to simulate embryonic development using an ECA paradigm that recapitulates the core task: independent subunits (whether cells or larger structures) working toward a species-specific endpoint. The "majority problem" is a task in which given an initial configuration of 0 s and 1 s, the cells of the ECA update their states at each timestep eventually converging on a final state where all cells have the same value—either 0 or 1—matching the value in the majority at initiation (Fig. 4a and Supplementary Fig. 3)[82–84]. Conceptually, we equate this to normal embryogenesis, in which a healthy embryo starts in a given configuration and develops towards the correct target morphology. However, what update rule is suitable to follow to achieve this goal state? In previous ECA studies, one update rule that solves this task is the GKL rule[85], which dictates that a cell's value at each timestep is set to match the value shared by the majority of a set that includes itself and defined neighbors to the left or right (see Methods). Not only does the GKL rule solve this problem, but it also does so efficiently, defined as converging to the correct value before a number of steps equal to half of the total number of cells.

Teratogens cause defective development or death by disrupting the endogenous, exquisitely coordinated cellular processes (such as sensing and information processing) that are required for morphogenetic coordination. Thus, we chose to model teratogenic perturbations in development as noise introduced as an alternate update rule, causing cells to mimic a single neighbor's state at random rather than following the GKL rule (see Supplementary Fig. 3). This noise is sufficient to cause dysfunction in how ECAs develop over time, resulting in incomplete or incorrect solutions at the end of the simulation (Supplementary Fig. 3).

To model conspecific interactions between embryos, we introduced a communication paradigm that relies on local communication in groups, similar to the local interaction between cells *within* ECAs (described above), but now at the level of interaction *between* ECAs. First, ECAs (representing embryos in a cohort) were given a spatial location on a 2D grid (i.e., growing embryos in a square configuration). Each 'embryo' was assigned a health value, which can vary between 1 (perfectly healthy) and 0 (dead) but was set at 1 for all 'embryos' at the start of the simulation (Supplementary Fig. 4). To model teratogen exposure, there is a chance (parameterized to 80%) that each embryo may be affected by noise (representing the deleterious effects of the reagent on the accurate functioning of the cellular machinery), in which case the embryo's health value is decreased by a percentage (parameterized to 70%). At each timestep, embryos that have been 'exposed' in this timestep or in previous timesteps broadcast their health value to their immediate neighbors and to their neighbor's neighbors (via a more diffuse and weaker signal; see Fig. 4 and "Methods" for details). This neighbor's neighbors' interaction proved to be a vital ingredient, as when the model was tested without it, survival plummeted for the same parameterization (Supplementary Fig. 5A). As could be expected for a biological stress signal, an 'embryo' in the model broadcasts if and only if it has been exposed. To simulate CEMA, at the next timestep (and until the end of the simulation), each embryo that received the health/stress signal shares a supportive signal in response. This supportive response scales with the cell's health value and, modeling our experimental finding that only exposed embryos can provide the CEMA effect, only 'exposed embryos' send a supportive signal. Incoming supportive signals are integrated as a weighted average. Additionally, we introduced a mechanism in which the healthier an embryo is (the closer its health status is to 1), the less it weights its neighbors' input and increases its own signal's weight; conversely, the unhealthier an embryo is, the more it weights its neighbor's input and decreases its own weight. Importantly, the model is designed to ensure that an embryo has no 'knowledge' of its neighbors' internal processes (we do not assume that embryos can sense the ground truth of the internal states of neighboring embryos). Instead, embryos can only send local signals that support (or destroy if the cell is noisy) update processes.

Based on the experimental data presented above, our hypothesis is that inter-embryonic signaling is a key mechanism in the collective robustness of morphogenesis. We thus simulated the development of embryos as singletons or in groups, similar to how embryos were grown in small or large groups. The results (Fig. 4) indicate that inter-embryo communication is essential for normal development in the presence of noise and that large groups are much more effective at achieving normal development than smaller groups, matching the embryo data in Fig. 1c. Overall, our agent-based simulation demonstrates that in morphogenetic systems in which emergent outcomes arise from local rules, collective dynamics can help stabilize against perturbation and noise (uncertainty) through local signaling mechanisms between agents.

## RNAseq analysis reveals the transcriptional signature of CEMA

Morphogenesis is the result of numerous biochemical, bioelectrical, and biomechanical signals, all of which feed into transcriptional

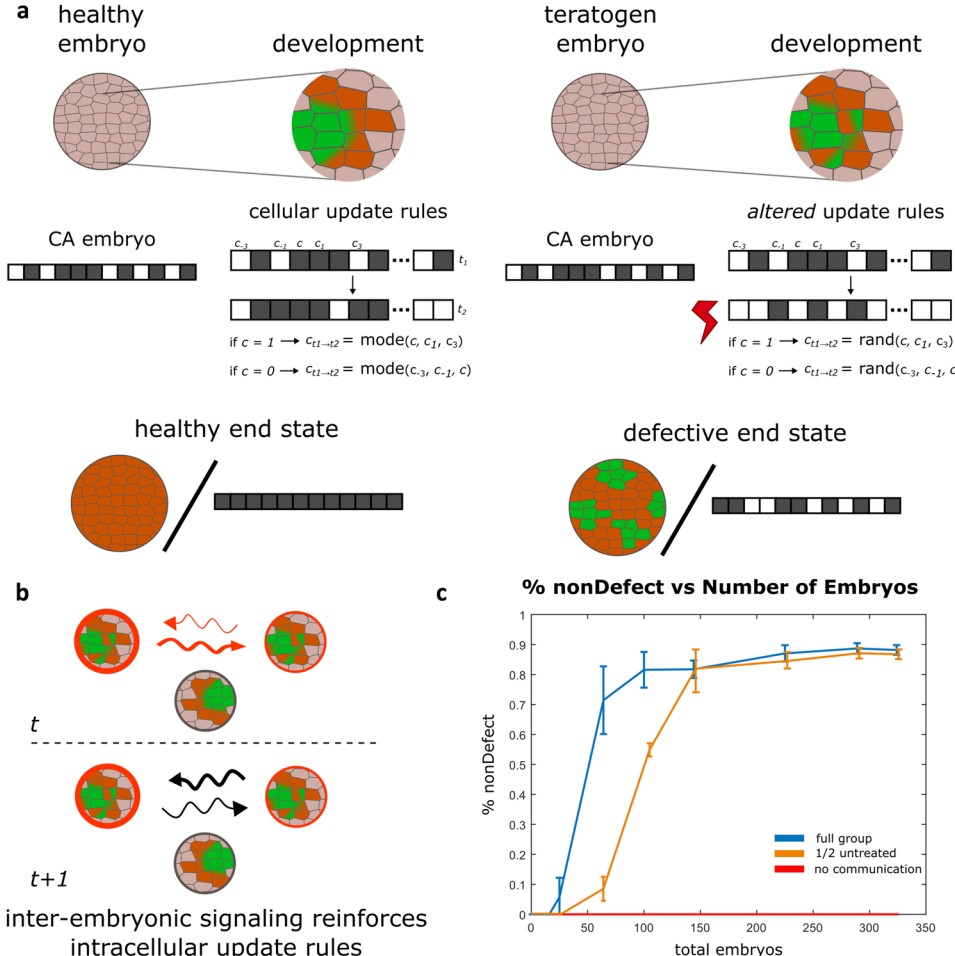

**Fig. 4 | Computational agent-based model of CEMA. a** Cartoon example of healthy and teratogen-influenced development. Left, an ECA, a 1D array of 0's and 1's, has an initial configuration of 40% 0's (white) and 60% 1's (black). ECA follows the GKL update rule, solving the majority problem by converging to all 1's. Right, an ECA that has noise induced by teratogens, which does not successfully solve the majority problem. **b** A cartoon example of inter-embryonic signaling from time $t$ to time $t+1$. Embryos signal their current health (red) at time t and supportive signals (black) at the next timestep to their neighbors and nearest neighbors. Embryos that have not been exposed to the teratogen cannot participate. **c** Data from the simulation shows that inter-embryonic signaling, CEMA, aids in development in the presence of noise (blue line). Without this communication, development fails in the presence of noise (red line). When half of a cohort is comprised of untreated embryos, they do not participate, and therefore the effect is lower (orange line). Each experiment included the number of embryos equal to the total embryos indicated on the $x$-axis and each experiment was repeated 50 times. The bars on the graph are 95% confidence intervals and the center represents the mean. Source data are provided as a Source Data file.

changes that canalize physiological events into cell differentiation and other long-acting morphogenetic modules. We next asked to what extent the CEMA effect involves changes at the transcriptional level: does gene expression reflect the differential responses to challenges that we observe in small vs. large groups? We performed RNA sequencing (RNA-seq) on *Xenopus* samples derived from large (300-embryo) and small (100-embryo) groups of conspecifics that were either untreated or exposed to thioridazine. A randomly selected group of 15 embryos were collected from each group ($N = 3$ samples/group) either immediately after thioridazine treatment (developmental stage 25) or after brief recovery in MMR media (developmental stage 35). At stage 25, we observed weak separation across groups when batch-corrected data was assessed by principal component analysis, with particular overlap across the untreated group with a large number of conspecifics and both thioridazine groups (Supplementary Fig. 6). Comparison of samples between the large vs. small conspecific groups by differential expression analysis found that no genes passed FDR-correction for either the control naïve or thioridazine-treated embryos. We conclude that by stage 25, the mechanisms implementing CEMA have not had a significant impact on gene expression profiles.

However, at stage 35, transcriptional separation was observed between differently sized and treated groups (Supplementary Fig. 6). Differential expression analysis revealed 32 significantly changed genes (FDR < 0.05) between embryos housed in untreated control groups of 300 vs. 100 in control media, with 30 genes increased and 2 genes decreased in the larger (300-embryo) group compared to the smaller group (Fig. 5a and Supplementary Table 1). This reveals that even in the absence of an external morphogenetic stressor, gene expression is sensitive to the size of the developmental cohorts.

For embryos exposed to thioridazine, 19 unique genes were differentially expressed between embryos housed in groups of 300 vs. 100, with 3 genes increased and 16 genes decreased in the larger (300-embryo) group (Fig. 5b), revealing that the presence of large numbers of conspecifics modifies the transcriptional response to teratogen exposure. It must be emphasized that our analysis is not simply picking up conventional transcriptional responses to a teratogen: the data in Fig. 6 reflect differential responses to the same insult in large vs. small cohorts. Thus, distinct transcriptional signatures exist for both conditions: simply developing in a large group (with no exogenous stressors) induced changes in gene expression, as did specifically being in a large group responding to a teratogen.

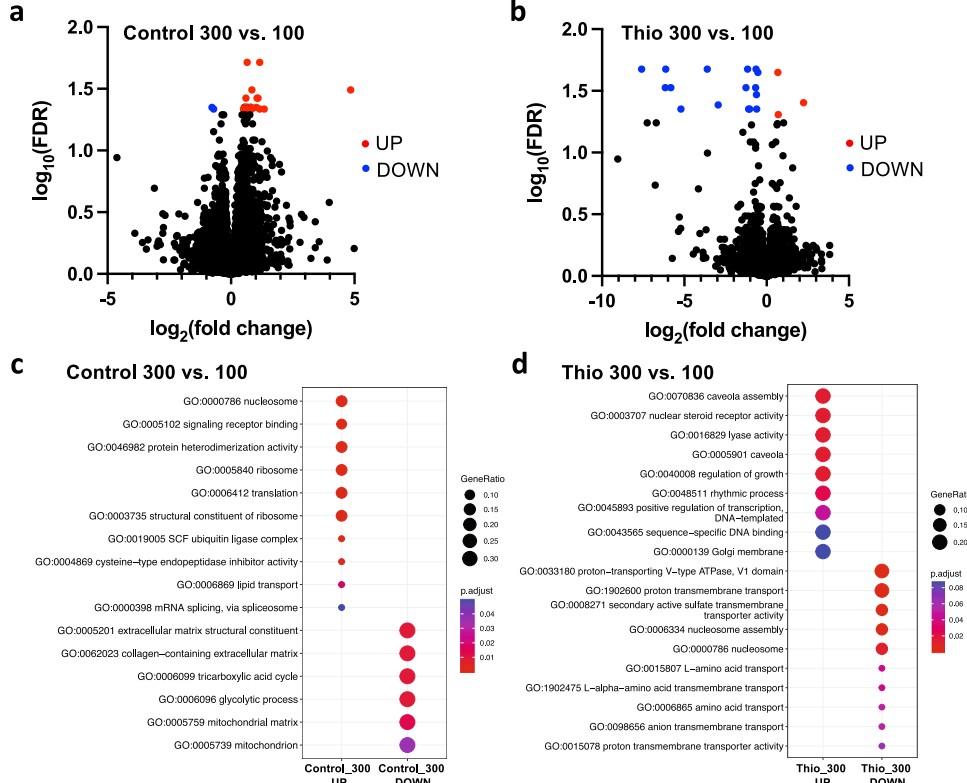

**Fig. 5 | Increasing cohort size induced changes to the transcription of different small sets of genes in control and thioridazine-treated embryos.** Volcano plots of differential expression data for large (300 embryos) compared to small (100 embryos) groups of **a** control and **b** thioridazine-treated embryos. Genes undergoing significant changes in expression (FDR < 0.05) are highlighted in red (increased expression) and blue (decreased expression). Pathway analysis for comparison of large vs. small numbers of conspecifics in the **c** control and **d** thioridazine treatment conditions. Pathway analysis was performed by over-representation testing of the differentially expressed genes (FDR < 0.1) in the gene ontology (GO) terms/pathways and over-represented pathways were identified at a significance threshold of FDR < 0.1. The size of the dots reflects the gene ratios (number of significant genes associated with the GO term/total number of significant genes associated with any GO term), and the adjusted p-value (FDR) reflects the significance. Source data are provided as a Source Data file.

We computationally identified biological structures and pathways potentially impacted by these transcriptional changes through over-representation testing amongst highly expressed genes (FDR < 0.1). In the control group, pathways related to nucleosome activity, signaling receptor binding, translation, and ribosomal pathways were over-expressed (FDR < 0.1) in the 300-embryo group compared to the 100-embryo group, while those related to extracellular matrix, glycolytic, and mitochondrial processes decreased (Fig. 5c). In thioridazine-exposed embryos, the expression of genes associated with signaling pathways, including caveolae and nuclear steroid receptors, was increased in embryos housed in groups of 300 vs. 100 (Fig. 5d). In contrast, genes involved in multiple membrane transport pathways were decreased in the larger group, including proton transmembrane transport. Together, these findings suggest that embryonic transcriptional programs are sensitive to the number of individuals in a cohort. In addition, the regulation of a small, diverse set of genes involved in multiple modes of signaling, between stages 25 and 35, is a signature of group resistance to teratogens.

## A diffusible molecule as a mode of communication

We next performed several functional blocking experiments to identify the modality responsible for the CEMA effect. First, to evaluate whether physical contact between embryos is required, 3D-printed separation devices were made. A clear, transparent plastic was used to create a platform with individual wells, measuring 1.80 mm tall and 1.80 mm wide, in which embryos could be placed (Fig. 6a). The entire separating device was placed inside of a dish thus allowing embryos to be physically separated, but still share media. A large (n = 300) group of

embryos was separated into individual solid, clear plastic wells while treated in thioridazine. While the wells were separate, the medium could diffuse freely across the top of the wells. Separating the embryos dropped the survival rate to 1.3% compared to 96% in groups where embryos were not separated from each other (Fig. 6b) (N = 3, p < 0.001 using Welch's t-test). To increase diffusion in the separation devices, new constructs were designed so that the walls of each well had windows that would not hinder diffusion. Animals treated with thioridazine in these new wells had an increased survival rate (63%) compared to the solid well-separated animals (1.3%) but were still lower than that for unseparated animals (96%) (Fig. 7b) (N = 3, p = 0.0005 using ANOVA). While testing for physical touch, optical cues were also tested through the use of the same separation device, but made from an opaque black plastic that prevented the sharing of optical cues. The results of blocking vision were not significantly different from blocking physical contact alone. Embryos left unseparated had an average survival of 93.7% while those treated in clear plastic had an average survival rate of 61.7% and 58.0% from those that were visually isolated (Fig. 6c). Based on these data, we concluded that the most likely medium of inter-embryo influence is a short-range chemical signal.

## Interfering with calcium and P2 receptors blocks the CEMA effect

To help identify the molecular nature of the diffusible signal, we focused on ATP, which is a known signaling molecule implicated in morphogenetic coordination in a range of model systems[86-88], as well as mediating long-range contraction responses after injury[51]. It is also rapidly diffusible, which makes it a plausible target given all of the

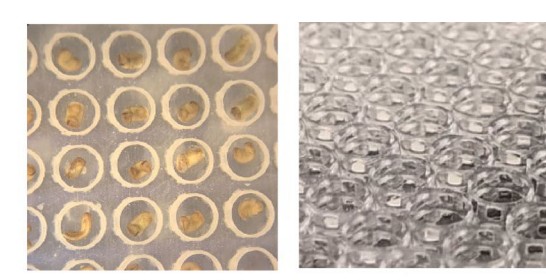

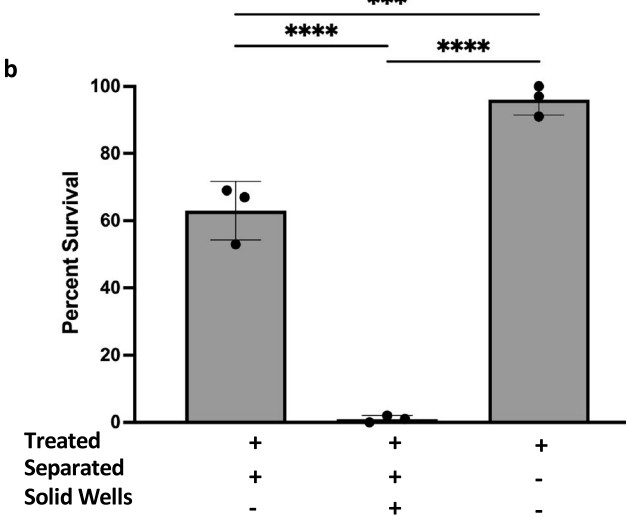

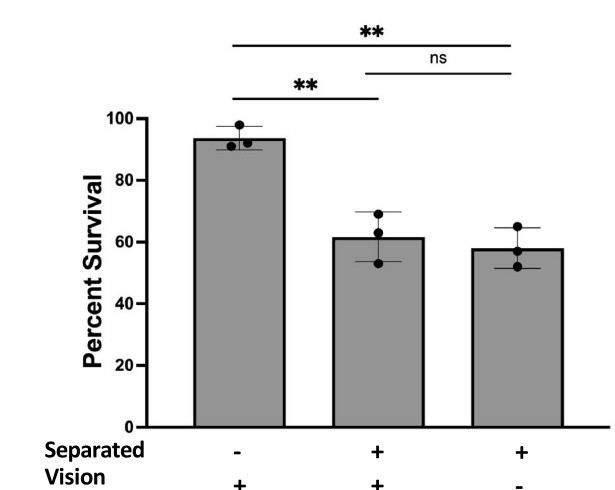

**Fig. 6 | CEMA requires diffusion but not physical contact. a** Images of embryos undergoing thioridazine treatment in a standard physical isolation device (left) and an image of a diffusion-enhanced physical isolation device (right). **b** Groups of $n = 300$ embryos were treated with thioridazine and were isolated using windowed wells, solid wells, or unseparated. ****$p < 0.0001$ and ***$p = 0.0010$. **c** Groups of $n = 300$ embryos were either put into clear or opaque separation devices or left unseparated and treated with thioridazine. **$p = 0.0021$ between -separated,+vision and +separated,+vision and **$p = 0.0012$ between -separated,+vision and +separated,-vision. **b**, **c** were both analyzed using one-way ANOVA and Tukey tests. Values are means ± SD. All treatments are normalized to the mean of non-treated controls' spontaneous defects. Source data are provided as a Source Data file.

above data. Studies have shown calcium and ATP acting in a range of different communication pathways, including injury response and intercellular communication[89–94]. Calcium waves in particular have been shown to act as communicators for damage and regeneration[93,95–97], suggesting these as potential candidates for inter-embryo signaling mechanisms for CEMA. We made use of a loss-of-function reagent known to block ATP signaling via the P2X/P2Y receptors: the purinergic receptor type 2 antagonist suramin[98–101] to see whether it would prevent the CEMA effect and render large groups as susceptible to defects as small ones[93,94,102].

Consistent with our results above, animals treated with thioridazine as singletons had a 96.2% incidence of defects; treating embryos in groups of 300 reduced this to 38.8%. Including suramin with thioridazine in a large group (300) treatments resulted in an incidence of 86.4%, not significantly different from the rate for embryos treated as singletons, suggesting suramin blocked the protective effect of the large group. Suramin by itself induced some defects in embryos treated in groups of 300, but only in an average of 7% of embryos (Fig. 7a, b) ($N = 3$). In an effort to identify a more precise molecular target, we focused on suramin's two major targets: calcium and P2 receptors. Previous studies have shown that PPADS blocks a number of different metabotropic and ionotropic receptors, including the P2 receptors[103,104], and BAPTA was used as a calcium chelator to investigate calcium's role. When thioridazine was used in conjunction with BAPTA, the average survival rate of a group of 100 embryos was 80.3% which was not significantly different from thioridazine alone, 86.7%. Using PPADS with thioridazine also caused a drop of average survival to 20.7% (Fig. 7c). These data indicate that calcium stores and P2 receptors are involved in the CEMA effect. To further investigate whether ATP is involved, we measured extracellular ATP levels with the hypothesis that larger groups expel more ATP into the media that signal to other embryos. Media from treated and control animals were harvested, and an ATP determination kit was used to quantify concentrations in the two conditions. Ratios of ATP concentration of the large group over the small group revealed that the average extracellular ATP concentration in treated groups was 1.073 µM while control groups had an average of 0.989 µM (Fig. 7d). This increase further supports a potential ATP-based mechanism for CEMA.

## Mechanical damage in *Xenopus* induces intracellular calcium waves in distant, uninjured conspecifics

We next sought to directly visualize the physiological dynamics that could underlie coordination between embryos using an acute mechanical injury assay. We reasoned that we may be able to observe such dynamics in the embryonic collective, and made use of a genetically encoded fluorescent calcium reporter, GCAMP6S. To directly visualize if *Xenopus* are sensitive to the damage of conspecifics, we first measured calcium signaling dynamics in GCAMP6S-expressing embryos exposed to mechanical injury as well as calcium activity in distant, uninjured embryos that share the same pool of 0.1x MMR media. We chose mechanical injury because it enabled a precise, known time point relative to which we could examine embryonic neighbors' activity. Experiments were conducted in custom channels to ensure repeatable media volume and embryo spacing paradigms (Fig. 8a). Mechanical damage is well-known to induce calcium waves within a single organism that aid in cell-cell coordination and permit rapid wound closure in *Xenopus* embryos in response to superficial injury[49,50].

Prior to injury, a low level of spontaneous flashes of calcium activity was detected. These spontaneous activity levels were quantified over 10 min of recording and used as a normalization factor for injury experiment quantification. Mechanical injury by a pulled glass

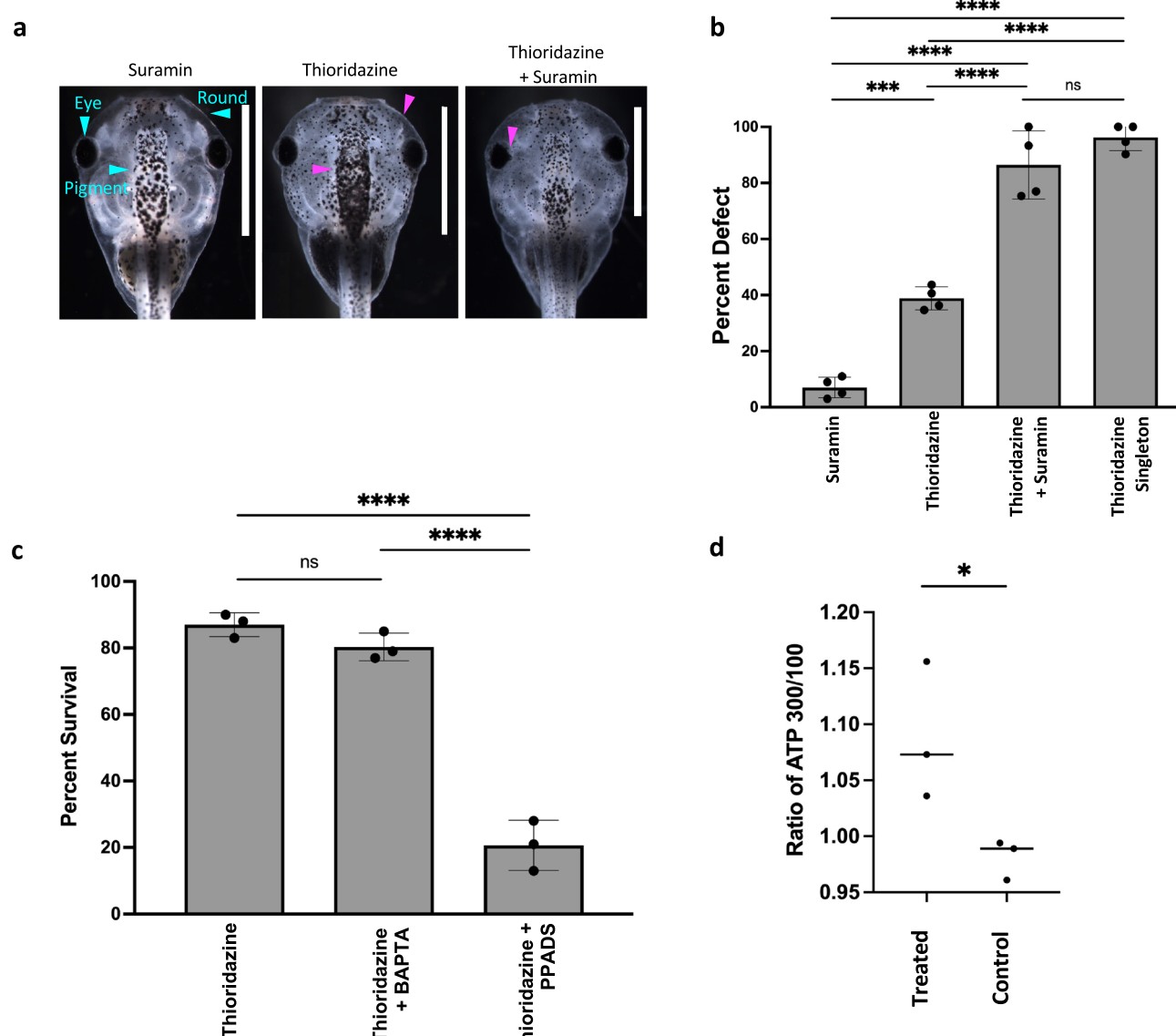

**Fig. 7 | ATP/P2X receptors may mediate CEMA. a** Suramin eliminates the protective effect of large group size against thioridazine-induced craniofacial defects and death. Tadpoles in groups of 300 were treated with 100 μM suramin, 90 μM thioridazine, 90 μM thioridazine and 100 μM suramin together, and compared to 90 μM thioridazine exposure as singletons. Blue arrows showing normal eyes, pigmentation, and head shape. Purple arrows show defects in those regions. **b** Percent defect of each group ($n = 300$ for each group) was then quantified. ****$p < 0.0001$ and ***$p = 0.0002$. **c** Thioridazine concentration was constant at 90 μM across all treatments and calcium or P2 receptor blockers were added. Calcium was depleted using 5 μM BAPTA. P2 receptors were inhibited with 100 μM PPADS. Each treatment group had a total of n = 300 embryos. ****$p < 0.0001$. **d** Quantification of extracellular ATP concentration. *$p = 0.0443$. Each dot represents the average of a group of three replicates. All animal images are dorsal views, and are oriented with the anterior end at the top of the image. For all data: values are plotted as mean ± SD, one-way ANOVA, and Tukey tests were conducted, and ***$p \leq 0.001$, **$p \leq 0.01$, *$p \leq 0.05$. All treatments are normalized to the mean of non-treated controls' spontaneous defects. Scale bars represent 2 mm. Source data are provided as a Source Data file.

needle induced a robust cell-to-cell calcium wave within the injured embryo (Supplementary Movie 1) and injury dynamics were recorded for 20 min post-injury. We observed that the injury wave was transferred to the second embryo sharing the same channel (Fig. 8b and Supplementary Movie 2), which was not itself manipulated in any way. Furthermore, we observed multiple injury wave transfers when an individual was injured within a group of 10 embryos (Supplementary Movie 3).

We measured cell-to-cell injury wave speeds of 6.7±3.48 μm/s in the injured embryo ($N = 7$), 5.28±1.89 μm/s between embryos ($N = 8$), and 2.36±1.66 μm/s in the uninjured embryo ($N = 9$). The wave speed in the uninjured embryo was slower than both the inter-embryo ($p = 0.023$) and injured embryo ($p = 0.045$) speeds (Fig. 8c). To more easily visualize calcium dynamics over time, the microscopy videos of

the uninjured embryo were transformed into kymographs depicting the pre-injury spontaneous activity and post-injury activity (Fig. 8d) and intensity vs. time curves (Supplementary Fig. 7). For untreated receiver embryos, post-injury calcium activity was detected within 5–10 min following mechanical damage in the adjacent injured embryo. Calcium dynamics in the receiver embryo included a combination of sporadic bursting activity as well as lower, background levels of consistent calcium activity. Across replicates ($N = 10$), the median peak of calcium activity in the injured embryo was 1.77-fold higher than spontaneous activity levels ($p < 0.0001$) and 1.08-fold change over spontaneous activity in the uninjured embryo ($p = 0.002$). Furthermore, the injury wave crossed distances of >1 mm in the medium, suggesting that diffusible molecules may allow for long-range signaling and detection of tissue damage between conspecifics.

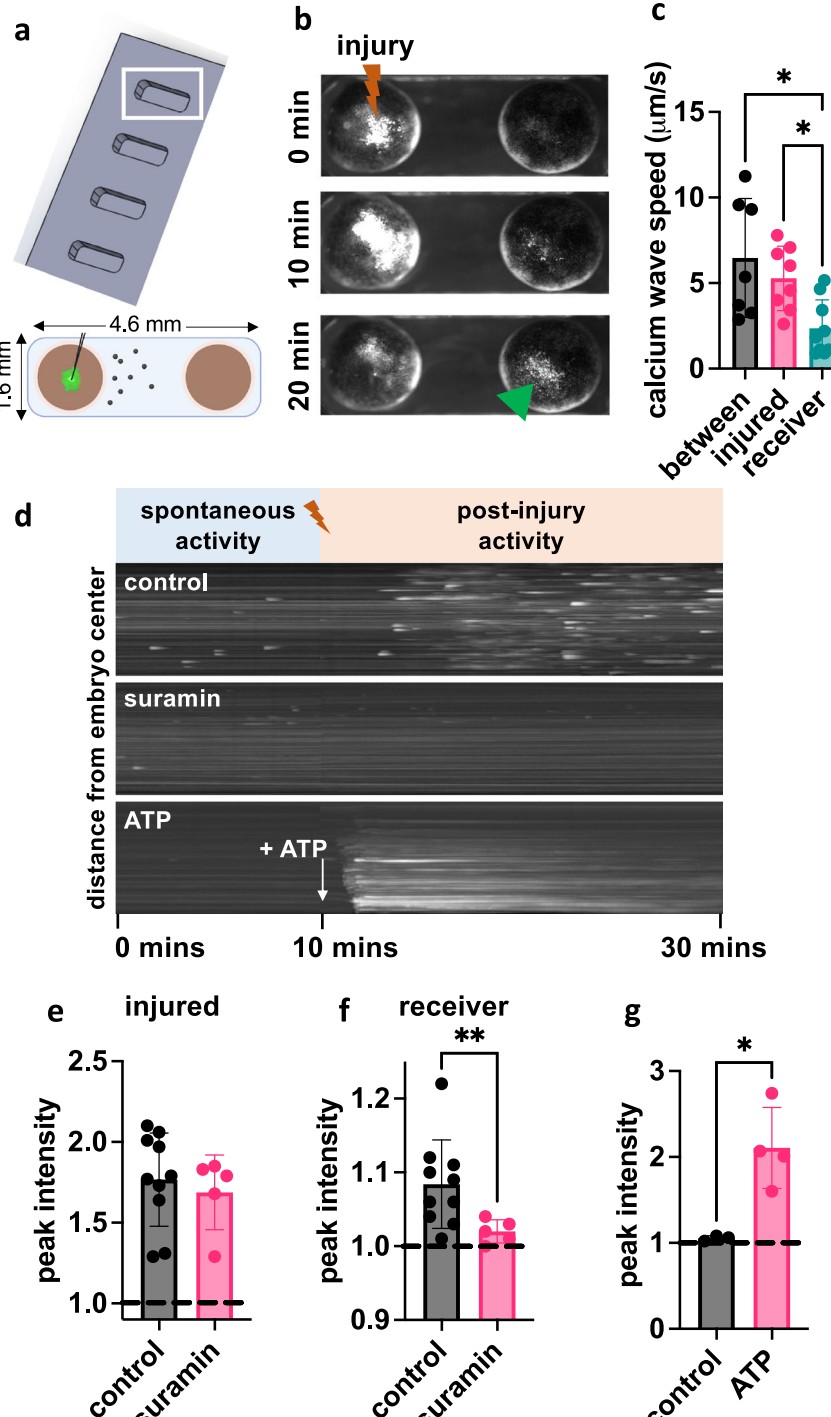

**Fig. 8 | Mechanical damage in Xenopus induces calcium waves in distant, uninjured conspecifics. a** Dimensions for holder and embryo arrangement for injury experiments. **b** Fluorescent images of GCAMP6S-expressing embryos at 0-, 10-, and 20-min post-injury for untreated stage 10–12 embryos. The injured embryo is on the left and the neighboring uninjured embryo is on the right. **c** The speeds of GCAMP6S signal propagation between and within embryos. Speeds were compared using the Kruskal–Wallis test, assessing differences in speed between embryos ($n = 7$) and within injured ($n = 8$) and receiver ($n = 9$) embryos; *$p < 0.05$. **d** Kymographs of receiver embryo calcium dynamics before injury (spontaneous activity) and post-injury under control, suramin, and ATP conditions. Maximum GCAMP6S signal for **e** injured and **f** receiver embryos from the control ($n = 10$) and suramin-treated ($n = 5$) groups. Intensities are normalized to the mean of the pre-injury spontaneous activity. Groups were compared separately for injured and receiver embryos using the two-tailed Welch's $t$-test; **$p = 0.009$. **g** Maximum GCAMP6S signal for embryos exposed to a bolus of MMR media (control; $n = 3$) or ATP ($n = 4$). Groups were compared using the two-tailed Welch's $t$-test; *$p = 0.01$. All error bars represent the standard deviation. Source data are provided as a Source Data file.

## Suramin treatment attenuates injury waves between embryos

We then tested if inter-embryo injury wave propagation uses signaling mechanisms necessary for CEMA by testing the P2-purinoceptor antagonist suramin, which interferes with ATP signaling. Embryos treated with suramin ($N = 5$) were tested in the injury wave assay and compared to a compiled set of controls ($N = 10$). No treatment suppressed GCAMP6S intensity in injured embryos below that of controls (Fig. 8e). Calcium waves traveled from the epicenter of injury to the

edge of the embryo as observed in controls. However, injury wave transfer to adjacent embryos was attenuated compared to untreated embryos in the case of suramin treatment ($p = 0.009$; Fig. 8e, f), suggesting that suramin suppresses only the *propagation* of the wave from the injured embryos to adjacent embryos. The mean peak of calcium activity was 1.08-fold change over spontaneous activity levels in the uninjured embryos in control media and 1.02-fold change over spontaneous activity in uninjured embryos treated with suramin (Fig. 8f). As suramin is defined as a P2 receptor antagonist, this finding suggests that injury signaling between conspecifics could involve ATP signaling. Indeed, when we apply a bolus of ATP, GCAMP6S is increased ($p = 0.01$) in uninjured embryos (2.11-fold change from spontaneous) over embryos exposed to a bolus of control media (1.05-fold change from spontaneous) (Fig. 8d, g and Supplementary Movie 4). However, the representation of the GCAMP6S signal in a kymograph (Fig. 8d) shows that the calcium *dynamics* are substantially different between the mechanical injury scenario and the application of a bolus of ATP. This difference in dynamics may occur because the mechanically injured embryo continuously releases diffusible molecules, including ATP, into the shared media pool, and the receiver embryo detects these molecules, resulting in prolonged calcium bursting behavior. When a bolus of ATP is applied in the media, the receiver embryo detects all the molecules in bulk, resulting in a strong, but short-lived, signal. Together, these differences suggest that simply applying ATP to media may not mimic the more complex signaling dynamics provided by nearby conspecifics that could provide an additional layer of information apart from the molecule itself. Additionally, because suramin has been shown to exhibit multiple off-target effects beyond its defined P2-purinoceptor antagonism[105], it is possible that additional molecules beyond ATP contribute to inter-embryo injury wave propagation and ongoing work in our lab aims to fully characterize the diffusible molecules involved in the injury wave effect as well as CEMA.

## Discussion

It is clear that although the hardware components of embryogenesis are encoded by the genome, phenotypic plasticity integrates inputs from the environment[11,21,22,106]. Here, we addressed the hypothesis that embryonic outcomes result not only from the activities of a single embryo's developmental mechanisms but also from a kind of collective computation[82,107–112], with stability properties arising from inter-embryo influence. Our results identify a source of lateral influence on development and morphology which resembles molecular- and cell-level collective repair[14,113].

Increasing cohort size (while scaling the amount of media and drug to be constant per embryo) was sufficient to mitigate the effects of different teratogens, including thioridazine, nicotine, and forskolin (Figs. 1 and 2). As cohort size increased, survival increased and the incidence of defects decreased (Fig. 1c, e), an effect we term Cross-Embryo Morphogenetic Assistance (CEMA). The observation of this protective effect of cohort size against defects induced by thioridazine, forskolin, and nicotine[57,114] (Fig. 2b, d), indicates that the effect is not specific to disruptors of a specific pathway. To further understand the boundaries of this assistive effect, we also looked at the effect of cohort size on defects known to be induced by over-expression of a dominant mutant of Kir6.1, a transmembrane potassium channel critical for bioelectric signaling during morphogenesis[56]. The protective effect of cohort size held for this disruptor as well (Fig. 2f). Finally, we also show that genetic diversity among cohorts does not affect the magnitude of CEMA and that CEMA is evident in perturbed rearing conditions, where larger treated cohorts have reduced baseline incidence of defects (Fig. 3). We conclude that individual *Xenopus laevis* embryos communicate with each other via instructive signals that protect against teratogenic insults.

These results, particularly the broad effect of CEMA across diverse disruptors and in control conditions, support the idea that robustness

in morphogenesis is partly the result of lateral interactions between conspecifics. This has precedence in several related biological systems. Perhaps the earliest (pre-metazoan) involves quorum sensing and coordination in bacteria[115,116]. Others include clutch hatching synchronization in response to perceived predation in birds[31,117], body color and brain changes in locusts depending on presence of others[118,119], and growth pattern of plant roots in order to avoid other plants or forage[120–122]. The boundary between the body and the outside world shifts on both ontogenic (developmental) and phylogenic (evolutionary) timescales[16]. Thus, it is perhaps not surprising that a kind of multi-scale dynamic is in play in morphogenetic homeostasis: interactions between cells, known to be crucial for normal embryogenesis, have a parallel in the interaction between embryos in a cohort[14].

### Molecular components of CEMA

We also gained insight into the mechanism of inter-embryo communication. Evidence of inter-individual communication has been noted in a variety of cases, including quorum sensing[123], viral lysis-lysogeny[124], and regeneration[125]. In an effort to understand the modality of communication in CEMA, special culture devices were used to physically separate embryos. Results indicated that CEMA required adjacency but not physical contact (Fig. 6), identifying diffusible chemical(s) as a likely inter-embryo signaling mechanism underlying CEMA. When embryos were physically distanced and prevented from touching, the average survival rate decreased significantly from non-separated, densely situated individuals. While separating embryos influenced survival, visual cues did not have any significant impact on the outcome.

Previous studies have found that following damage, ATP is released in high concentrations into extracellular space and can induce calcium waves[126–129]. In our study, we have shown that increasing the number of embryos in a group has a protective effect when faced with a stressor (chemical or physical) and that either depleting calcium or blocking P2 receptors can eliminate this shielding effect (Fig. 7).

The involvement of P2 receptors in craniofacial development is not surprising as purinergic receptors are found in nearly every tissue in adult animals[130]. In zebrafish, P2X3.1, a paralog of P2X3 mammalian receptors, is expressed in the embryonic head, specifically neural and ectodermal cells. Knockdown of P2X3.1 was shown to result in malformed lower jaws, malformed branchial arch regions, and smaller heads due to defective pharyngeal skeletal development[131]. It was also noted that the loss of signaling also disrupts normal neural crest behavior within the branchial arches. In relation to *Xenopus laevis*, P2 receptors have been detected in the central and peripheral nervous system[132]. The authors note that while each subunit has a distinct expression profile, there are several that are expressed in the head region. At stages 27 and 33/34, the p2rx1.L subunit is localized to a specific region of the hindbrain as well as in the head region, in the branchial arches, and in the cement gland. Expression of the p2rx1.S starts at the cleavage stage and then localizes to the eye field during neurulation till stage 41. The expression of p2rx2.L is mostly found in developing mesoderm derivatives and the nervous system. Expression can be seen in the head region, the brain, and sensory organs like the eyes at stage 41. Finally, a third member of the p2x receptor family is weakly found at the somites (stage 33/34), p2rx6.L. At stage 41, this receptor expression is strong in the head and dorsal regions. Temporally, the start of expression may be unique, but the level of expression for all of them increases during development and reaches a peak at stage 45.

When embryos are physically injured, they elicit a calcium wave that starts at the site of injury, propagates across the embryo, and spreads to a neighboring unharmed embryo (Fig. 8). It should be noted that, as always, the identification of the molecular components underlying the effect is just part of a full explanation. A simple concentration of ATP or any other molecule does not have the bandwidth to specify the information needed for complex morphogenesis as observed here. While it's formally possible that CEMA involves a

counter-teratogen effect rather than a pro-morphogenetic effect (e.g., a signal to degrade teratogenic drugs or mRNA, rather than a signal to improve morphogenesis), we think it unlikely based on the specificity of the effect (failure to cross-protect) and because evolutionarily, it would be much easier to develop mechanisms that support one particular target morphology vs. ones that try to anticipate the huge variety of possible teratogens. Future work will address the dynamic encoding of patterning cues via these molecular implementations, which could include spatial or temporal patterns (e.g., pulse-coding as has been observed with calcium[133–136]).

### Calcium waves as potential inter-embryo mediators of communication about injury/insult

Consistent with previous studies, we found that an intra-embryo calcium wave is induced when mechanical damage occurs[50,51]. Furthermore, we observed for the first time that the calcium wave not only travels across the injured embryo but also into and across adjacent, uninjured embryos (Fig. 8 and Supplementary Movies 1–4). This occurs at a velocity in the same order as that observed for cell-to-cell signaling in developing *Xenopus laevis* embryos (5 μm/s) and injured *Xenopus* neuroepithelium (9 μm/s)[93,137], but much slower than that observed for cardiac electrical conduction (2–5 m/s), neuronal action potentials (0.5–100 m/s), or cranial blood flow (24–100 cm/s)[138,139]. It is also slower than the long-range signaling observed between injured and uninjured limbs within 5 s of amputation in a model of hindlimb regeneration in *Xenopus*[125]. Further, targeting ATP signaling with suramin, an antagonist of purinergic receptors P2X and P2Y, abrogated both the inter-embryo calcium wave and the CEMA effect against morphogenesis disruptors (Figs. 7 and 8), suggesting a common ATP signaling mechanism. While these results do not directly implicate inter-embryo calcium waves in CEMA, they suggest that embryo damage can cause diffusible signaling molecules to be detected by nearby conspecifics. While a mechanical injury and chemical treatment both induce stress or injury onto the embryo, the methods and scale of doing so are quite different. The mechanical injury serves as an example of a short, discrete stressor, compared to a longer-term, continuous, teratogen-based stressor. The mechanical damage observed in the injury wave assay may induce different signaling responses than that induced by chemical injuries, like thioridazine and nicotine. By probing both modalities of trauma, we were able to examine whether there is a shared mechanism despite their differences. Additional modes of injury should be investigated in the injury wave assay, including possibly chemical and laser-induced damage. The above-implicated mechanisms are not meant to be exclusive, and it's entirely possible that other modalities are also involved.

### Transcriptional signature of CEMA

Communication between cells and embryos that affects morphogenetic outcomes is likely to involve downstream changes in the expression of developmental genes. In an effort to identify unique transcriptome features between large and small conspecific groups, an RNAseq analysis (using rRNA depletion, to capture more unique features) was done on samples from the thioridazine experiment. Our analyses showed that changes are first detectable between stages 25 and 35, and involve a small number of genes that specifically respond to cohort size and not simply to the action of a teratogen. There were 16 genes that were down-regulated and 3 that were up-regulated (Supplementary Table 1). Many of these are currently of unknown function, but the list includes the V-ATPase, which is well-known to drive morphogenetic events[140–146], and interesting new components linked to DNA damage and endopeptidase activity. Gene set enrichment analysis showed that genes associated with signaling pathways, including caveolae and nuclear steroid receptors, were increased in large groups, but genes involved in multiple membrane transport pathways were decreased in the larger group (Fig. 5). These changes

also distinguish the context of a cohort responding to a challenge vs. simply to the size of a developmental clutch (Fig. 5).

### A model for intra-embryo signaling about developmental disruption/injury

Taken together, the data suggest the following signaling elements (Fig. 9). We propose that in normal conditions, the extracellular concentration of ATP is low. Injury induces an internal calcium wave and increased ATP release from the injured embryo. The fact that suramin or PPADS both block the calcium wave in neighboring intact embryos and, blocking P2 receptors prevents a calcium response, suggests that ATP released from injured embryos binds to P2 receptors on the surface of neighboring embryos, triggering the secondary calcium response in uninjured embryos (Fig. 9).

### Interaction dynamics of CEMA: a computational minimal model

Much as in traditional developmental studies of single embryo morphogenesis, it is not enough to identify the molecular components necessary for CEMA. Previous work indicates that the robustness of normal craniofacial development is due to the computational capabilities of cell collectives, which allow them to reach appropriate target anatomy despite perturbations in the starting state or environment. To begin to identify system-level collective dynamics sufficient to produce group robustness across embryos, computational modeling can help interrogate the ways in which instructive lateral signals might protect embryos in large groups against disruptive signals. In addition, it can motivate intervention strategies. There are many examples of stability in groups, for example, coupled oscillators—from Huygens' clocks to networks of networks[147–150] or of robot swarms[151]; the "wisdom of crowds"[152] is now a well-known phenomenon.

We modeled this effect using agent-based, cellular automata, with an emphasis on anatomical homeostasis—the ability of morphogenetic systems to resist noise. In this model, embryos that are affected proportionally seek necessary morphogenetic signaling information from others in the cohort. Numerous examples exist of embryos sensing and progressively correcting for powerful disruptors to early developmental processes, such as those that affect left-right asymmetry[153], but to our knowledge this has not before been studied at the group level. This aspect of the model is a formalization of the idea that stress propagation is an important aspect of collective problem-solving because it leads to multiple subunits taking action to address the same unmet need. It is easy to imagine how this could have evolved: autocrine signaling can readily become paracrine signaling, at whatever scale of organization. Selection could progressively reward the leakiness of signals that represent the current error state as broadcasts that enable integrated activity to reduce the error.

Our model reinforced the conclusion that inter-embryonic signaling is a key factor in CEMA. A key component is that we modeled both embryonic development and CEMA using local interactions, not requiring a central controller mechanism. These types of local interactions are a hallmark of complex collectives that are found across biology and are hypothesized to support a number of functions[154,155]. In our simulations, local interactions were sufficient to solve the majority problem, overcoming noise and supporting 'normal development', with the assumption that the local interactions are further supported by the local cell's neighbors (neighbor's neighbors). We see that when these interactions are taken away, or reduced to a significant degree, the population becomes much less robust to perturbation (Supplementary Fig. 5). However, one open question in this study, as well as in the field of complex systems, is how many agents (i.e., embryos) do you need to become a 'collective' that functions differently than smaller groups? Here, the experimental and simulation data support a number around 300 for 100% survival in the presence of a teratogen but we do not have a clear picture as to *why this specific number* and if this number is unique to this species or can be

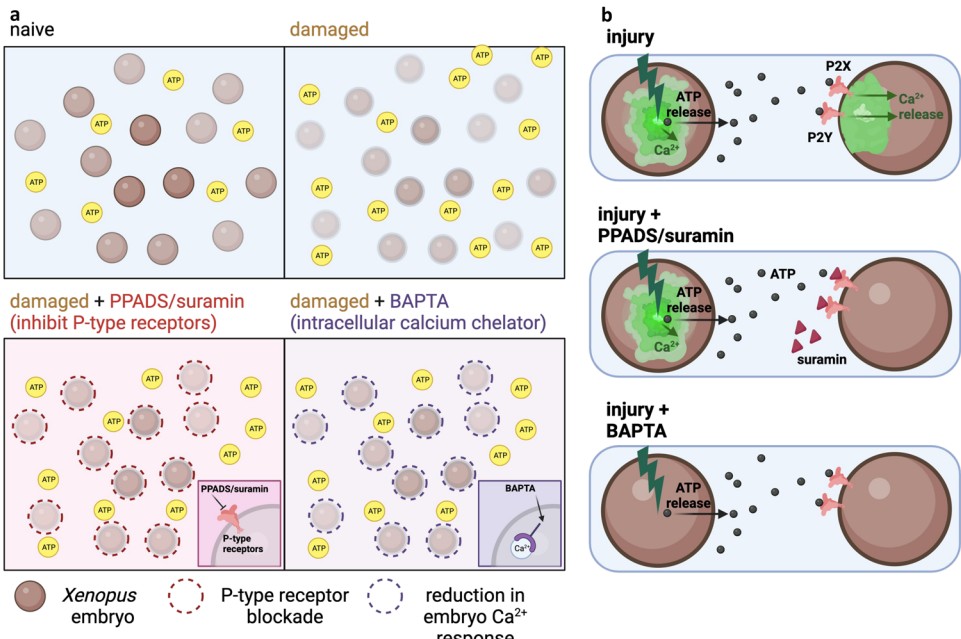

**Fig. 9 | Mechanistic model of CEMA effects. a** Naive condition in which embryos are undisturbed. Damaged conditions in which embryos are subjected to injury and ATP expulsion increases. Damaged embryo + PPADS/Suramin showing the environment in which P-type receptors are blocked and inhibiting the binding of ATP. Damaged + BAPTA shows that internal calcium stores are depleted and there is a reduction of calcium response. **b** Enlarged view of internal events during injury. Top: The embryo on the left is injured and elicits a calcium response and increased ATP release. ATP binds to P2X receptors on neighboring, uninjured embryos and elicits its own calcium response. Middle: In the presence of PPADS/Suramin, the injured embryo has a calcium and ATP response but P2X receptor blockade prevents ATP binding and subsequent calcium wave in the uninjured embryo. Bottom: In the presence of BAPTA, the injured embryo no longer has a calcium response, but is still able to increase the release of ATP. ATP can still bind to P2X receptors in the neighboring embryo but there is no calcium wave, indicating that the calcium responses in both the injured and uninjured neighbor are dependent on calcium. Created with BioRender.com.

generalized to other systems. In future studies, this "behavior vs. cohort size" phenomenon could be investigated if the relationship between them is similar. For example, in both the experiment and model we see a sharp, seemingly non-linear relationship between cohort size and survival.

While our results do reflect what is seen in Fig. 1c, they do not match perfectly. In our simulation groups of fewer than 25 conspecifics demonstrate small robustness against disruption by noise (<10% survival), whereas in the experimental data in Fig. 1c, groups of 5 and 10 embryos show no significant resistance to the effects of teratogen. Our model could be extended in several ways in future work. First, while there is embryonic death during the simulation, there is no 'death-signaling', wherein a dying embryo may emit a destructive signal that affects other embryos and could greatly affect development. Second, all ECAs are given a random initial configuration of 0's and 1's, with a majority (60%) set to 1. However, it may be that initial embryonic stages are not so random but have some amount of reproducible structure. Further experimental and computational studies are needed to address this issue. Third, while our inter-embryonic signaling is indeed local, it does not necessarily account for inter-embryonic traveling waves. If signals can propagate across a larger space over time in a relay fashion, this may change how the collective develops. We are currently developing a more in-depth model to address these issues in future work.

**Limitations of study and extension in future work**

A number of questions remain open. First, we do not know how this effect operates in the wild or how the many additional factors of a complex natural environment will impact CEMA. Thus, while we have shown that embryos *can* cooperate in groups to resist teratogenesis, we do not yet know how much this affects evolution and adaptive fitness in nature. We also do not know the limits of this effect across types of injury. We do not claim it is universal, and many more perturbations beyond the three tested above will need to be tested to see how broadly general this effect may be. It would be interesting to see what other types of developmental perturbations can cause signal generation and propagation. Subsequent experiments will look at gene expression changes in individual animals of a cohort, paralleling the strategy of doing single-cell RNAseq in single bodies[156–158], to help understand the scaling of instructive cues and their transcriptional responses.

Other avenues for future work are now open. This needs to be investigated in other systems, particularly mammals. It is already known that calcium waves exist across mammalian tissues[92,94], and that mammalian cells in culture do much better in groups than they do alone[159,160], but we don't know if this extends to cohorts in embryonic mammalian development. The general field of Allee effects[35–37] in novel embodiments is a fascinating area for future investigation, especially as it may connect with other instances of horizontal transfer of influence via substrates of different scales, from exosomes[161–163] to whole tissues or even organs transplanted across bodies[164–170]. We hope that the above dataset and unconventional assay form the basis for an integrated approach to understanding the robustness of collective decision-making across scales, adding to the growing literature on collective problem-solving[18–20,171–174].

There are also potential connections to evolutionary questions related to group selection[175–177], which may be enriched by a better understanding of what group dynamics contribute to the developmental fitness of each member of a community. It is likely that interesting evolutionary developmental biology analyses could emerge from a broad study of CEMA, impacting basic questions about the information flows that determine embryonic outcomes[178,179]. In general, these types of inputs into developmental outcomes stretch the concept of epigenetics via environmental influences. Moreover, cross-

embryo beneficial influences challenge the assumption that non-genetic external factors are always due to inanimate physical aspects of the environment, conspecific competition, or harmful exploitation by other biotic agents[180].

## Implications

Taken together, these data reveal an essential role for inter-embryo signaling during morphogenesis, adding to the growing body of work indicating that the relationship between genomic information and anatomical outcome is not as straightforward as expected. For example, planaria accumulate many mutations through somatic inheritance but are able to regrow any missing part with 100% fidelity to reach a target morphology[181]. The fact that organisms can reach their correct morphology despite variation in genetic material shows that there is a gap in understanding all the informative sources of morphology. It is essential to continue to elucidate diverse instructive inputs to understand the mechanisms of anatomical plasticity and robustness.

The discovery of the CEMA effect may have several practical impacts. The first concerns available data on the developmental toxicity of various agents. Our results indicate that the percentage of defects (i.e., teratogenic potential) identified in assays such as FETAX[182–185] are unwittingly adjusted for CEMA—the degree of teratogenicity reported is what is seen after an unknown level of embryonic assistance has taken place (since these assays are almost never done on singletons). In other words, because most studies do not compare (or sometimes even state) the size of the cohorts, we rarely know the actual effect of a given agent—only its effect after possible CEMA mechanisms have had a chance to improve it. This suggests that it is essential to state the cohort size in teratogenicity assays, and even to re-do many of the most important studies using different embryo group sizes, to improve the transferability of the findings to human embryos where the cohort size is much smaller than in Xenopus or Zebrafish models.

The more positive implication concerns how understanding CEMA might improve biomedicine. It is tempting to speculate that, having understood how a group of agents signals each other to establish a healthier outcome, we may someday be able to artificially trigger that response in therapeutic settings. While much more work needs to be done to properly evaluate the potential of this effect for biomedicine, it is clear that beneficial, instructive cross-embryo communication, understood broadly, is an exciting phenomenon that could shed light on evolutionary fitness of developmental mechanisms and could perhaps be hacked to address urgent biomedical needs in birth defects and other disorders of morphogenesis[186,187].

## Methods

Animal care was done in compliance with and approval from the Institutional Animal Care and Use Committee (IACUC) under protocol number M2023-18 of Tufts University.

### Animal husbandry

Xenopus laevis tadpoles were reared in 0.1× Marc's Modified Ringers solution (MMR), pH 7.8 using standard procedures[188] and were staged according to Neiuwkoop and Faber[189]. All embryos (pooled from separate mothers and then randomly divided between treatments) were raised at 14 °C. Feeding stage animals were fed 3 times a week with Sierra Micron powdered diet, and media changes were performed M, W, and F. Tadpoles were raised in either small dishes or large dishes depending on the cohort size.

### Pharmacological exposures

Thioridazine (thioridazine-HCl, Sigma-Aldrich) was dissolved in deionized water at 1 mM and frozen in single-use aliquots to prevent continuous thawing. Neurula stage embryos that had been reared at 14 °C prior to exposure were treated with 90 μM thioridazine in 0.1x MMR from NF stage 12.5 to 26. During the treatment, experimental and control embryos were kept at 18 °C. Post-exposure they were returned to 14 °C until they reached scoring stage 45. A ratio of 1 mL of media to 1 embryo was kept constant across group sizes (therefore 40 embryos were raised in 40 mL and 120 embryos were kept in 120 mL).

Forskolin (Forskolin, Tocris) was dissolved in dimethyl sulfoxide at 10 mM, aliquoted into single doses, and frozen until use. Stage 10 embryos were exposed to 5 μM forskolin in 0.1x MMR. Animals were reared at 14 °C and media was refreshed three times a week (M, W, F) until animals reached scoring stage 45.

Nicotine (Sigma-Aldrich) exposure occurred from stage 11 to stage 35 at 0.1 mg/mL nicotine in 0.1X MMR, with media/drug refreshed every other day. Animals were kept at 18 °C during treatment. Post-treatment, animals were rinsed with 0.1x MMR twice, moved into fresh 0.1x MMR, and kept at 14 °C until stage 45.

Suramin (Suramin sodium salt, Sigma-Aldrich) was dissolved in dimethyl sulfoxide at 10 mM, aliquoted into single doses, and frozen until use. Exposure occurred with 100 μM suramin in 0.1x MMR conducted on neurula stage embryos (NF stage 12.5–26) that had been reared at 14 °C prior to exposure. During the treatment, experimental and control embryos were kept at 18 °C. After the exposure, they were returned to 14 °C until they reached scoring stage 45. Calcium wave experiments were performed on pre-neurula stage embryos (NF stage 10–12). Embryos were exposed to 100 μM suramin in 0.1x MMR for 30 min prior to treatment at 18 °C. Embryos were maintained in suramin during the measurement of spontaneous and post-injury calcium activity.

### Morphometrics

Tadpoles used for morphometric analysis were imaged with a Nikon SMZ1500 microscope with a Retiga 2000R camera and Q-capture imaging software. Landmarks for morphometric analysis were based on reproducibility across tadpoles. ImageJ was used to measure the width between two points (head width). For head width, landmarks were the outermost points of the head (generally near the middle of the eyes), marking the diameter and base of the branchial arches.

### Microinjection

mRNA for a chimeric construct that is dominant negative for Kir6.1 was synthesized using standard message machine kits (Life Technologies) and stored at −80 °C until used. Embryos were transferred to 3% Ficoll solution before being microinjected. Pulled capillary needles were used with bubble pressure between 55–60 kPA and injection pressure set to 140 kPA. Injection time was set at 100 ms and at the 2-cell stage, 2 out of 2 cells were injected. Immediately after injection, embryos were moved to fresh 3% Ficoll plates and left to recover for 1 to 2 h. After that timeframe, half the media was poured out and filled with 0.1x MMR for another hour. Afterward, embryos were washed twice in 0.1x MMR and moved to a 14 °C incubator. Similarly, mRNA for the genetically encodable fluorescent calcium reporter GCAMP6S was synthesized from linearized template DNA by the message Sp6 in vitro transcription kit (Thermo Fisher) using previously described methods[190,191]. Roughly 2 nl of 300 ng/ml mRNA was delivered at the 4-cell stage to 4 out of 4 cells by microinjection.

### Injury induction and calcium imaging

GCAMP6S-injected embryos (NF stage 10–12) were loaded in groups of 2 into custom-machined acrylic holders with channels (4.6 × 1.6 × 2.5 mm). A thin layer of mineral oil (Sigma-Aldrich) was added on top of each channel to prevent drying during imaging sessions. Using a ZEISS Axio Zoom.V16 microscope and frame rate of 1 image acquired every 2 s, 10 min of spontaneous GCAMP6S activity was captured for all embryos at baseline. Localized mechanical injury was induced in 1 of the 2 embryos by a pulled glass capillary needle.

Using identical imaging parameters, GCaMP6s expression was captured for all embryos from the time of injury for a duration of 20 min.

In Fiji (ImageJ), image stacks including baseline and post-injury activity were cropped to separate image stacks for each embryo (injured and receiver) and each stack was registered using the descriptor-based time series registration plug-in. A circular region of interest was selected to encompass each embryo and the mean gray value was output for that region across all frames acquired. The mean gray values after injury were normalized to the baseline activity collected for each embryo.

The peak of activity for a given region was defined as the maximum normalized signal at any time during post-injury data capture. The distance between any 2 regions of interest was calculated by finding the magnitude of the vector from the injury site to the closest point on the receiver embryo. The time of calcium wave transfer between embryos was taken as the time from injury induction to the time that the mean calcium signal surpassed the maximum of spontaneous activity within an embryo. To avoid detecting aberrant spikes, a transfer between embryos was only considered complete if the mean calcium signal was maintained above spontaneous levels for longer than 1 min. The cell-to-cell speed within a single embryo (injured embryo & receiver embryo speeds) was measured by selecting a cell at the start of a calcium wave and a cell at the end of the calcium wave. The time from cell-to-cell was calculated by measuring the peak-to-peak time delay between each cell. Speed for intra- and inter-embryo measurements was defined as the time of calcium wave transfer divided by inter-region distance.

## CEMA computational model

All models were built using the MATLAB software (9.12.0.1884302, R2022a). Each ECA was created with 149 cells with a full runtime of $N_{cells}/2$, a common configuration[82,83]. To simulate development, ECAs were given the GKL rule to solve the majority problem, with an initial configuration of 60% 1s and 40% 0s[84]. An embryo was considered successfully developed with no defects if all cells had converged to 1 before the end of simulation time. To simulate embryo groups, they were developed in a square configuration ($2 \times 2$, $3 \times 3$, $10 \times 10$, for example) from values ranging from singletons (1 embryo) to large groups ($18 \times 18$, or 324 total embryos). Tested values were chosen to emulate the experimental data.

To simulate health and noise (teratogen), each embryo was given an initial health value of 1. At each timestep there is a chance that noise, $p_{noise}$, could affect any embryo in the group, altering its health by a percentage, $n_{dec}$, parameterized to 0.7. Then, each embryo affected by the noise will update its health value based on a weighted average of its neighbors and neighbor's neighbors. The embryo's weight is 1, its neighbor's weight is 1, and its neighbor's neighbors' weight is 0.25. These values were chosen to simulate spatially close mixing as well as diffuse, slightly distant signaling. This signaling was vital to the CEMA effect, as when the weights were turned low (<0.01) the collective never recovered unless noise was turned down to a substantial degree ($p_{noise} = 0.6$) (see Supplementary Fig. 4A). After, each embryo affected by noise will again update its health based on supportive signaling, sampled from the previous timestep. This health update is also calculated based on a weighted average, with the weights of the embryo's own signal proportional to *(1-health)*, as is the signal from its neighbors. The weights of the neighbor's neighbors are weighted as *0.25\*(1-health)*, again to simulate diffuse signaling. This weighted average is then applied to the embryo's health value following the formula:

health value + = current health value*weighted support signal

After this, an embryo then evolves according to the GKL rule if health is above 0.5, or the random update rule if health is equal to, or less than, 0.5. This threshold was chosen as the intermediate value

between 0 and 1, however, other values were tested (see Supplementary Fig. 4B). At higher values (0.75) embryos deteriorated too quickly for recovery, and for lower values (0.25), recovery became trivial, even for smaller conspecific groups. For parameterization, noise values from 0.0 to 1 were tested in steps of 0.1, and we found that noise of 0.8 best reflected the data found experimentally. For $n_{dec}$, parameterization found that 0.7 best fit the experimental data. Each simulation was run 20 times to create a sample space, and 99% confidence intervals were calculated for each noise value across a number of embryos.

To simulate the experimental design when embryos not affected by the teratogen were introduced into a group that had previously been exposed, we artificially locked half of the population to 1, meaning they stayed perfectly healthy. With this, they were not allowed to participate in the health signaling.

## RNA extraction

At stage 35, embryos were sacrificed for an rRNA depletion study. A sample consisted of 15 pooled tadpoles, and each was repeated 3 times. Tissue was extracted using TRIzol (Thermo Fisher Scientific) as per the manufacturer's protocol, and total RNA quality and quantity were assessed using a NanoDrop spectrophotometer (Thermo Fisher Scientific).

## rRNA depletion RNA-sequencing

RNA was sent to the Tufts Genomic Core where RNA quality was assessed via bioanalyzer, and high-quality RNA was used for library preparation with the Illumina Stranded Total RNA with Ribo-Zero Plus. Libraries were then multiplexed, and an rRNA depletion run using single-end, 75-nucleotide sequencing was performed on Illumina HiSeq 2500. Raw read files were sent to the Bioinformatics and Biostatistics Core at Joslin Diabetes Center.

## NGS analysis

Reads were trimmed for adapter "CTGTCTCTTATACACATCTC CGAGCCCACGAGAC" and polyX tails, then filtered by sequencing Phred quality (>=Q15) using fastp[192]. Adapter-trimmed reads were aligned to the *Xenopus laevis* rRNA genomic sequence (version 10.1) from the NCBI nucleotide database using Bowtie2[193] and unmapped reads were removed using samtools[194]. Adapter-trimmed reads were aligned to the genome using a STAR aligner with the two-pass option. The RSEM tool was used to estimate the gene expression from the gene alignments. Low-expressing genes (threshold of 1.8 counts per million in at least 3 samples) were filtered out, leaving a total of 21,720 genes after filtering. Counts were then normalized by the weighted trimmed mean of M-values (TMM)[195]. The counts were Voom transformed[196] into logCPM (*logCPM=log₂(10⁶\*count/(library size\*normalization factor))*). Two surrogate variables (SVs) were identified and constructed[197]. Batch and the SV effects were adjusted and a principal component analysis (PCA) was performed. Differential expression analysis between groups was performed using limma[198]. Pathway analysis was performed by testing the over-representation of the differentially expressed genes (FDR < 0.1) in the Gene ontology (GO) terms/pathways using R package clusterProfiler[199]. Enriched pathways were identified at a significance threshold of adjusted *p*-value (FDR) < 0.1. In the gene ontology dot plots, the size of the dots reflects the gene ratios (number of significant genes associated with the GO term/total number of significant genes associated with any GO term), and the adjusted *p*-value (FDR) reflects the significance.

## Statistics

All statistical analyses were performed using Prism 9. To achieve statistical power, biological replicates (*N*) were conducted 3–6 times with $n > 50$ embryos for each treatment unless otherwise noted. Data across various iterations were pooled and analyzed by non-parametric *t*-test (for two groups) or ANOVA (for more than two groups).

## Reporting summary

Further information on research design is available in the Nature Portfolio Reporting Summary linked to this article.

## Data availability

RNA-sequencing data generated during the study are available in the NCBI GEO public repository with accession no. GSE245782. Source data are provided in this paper.

## Code availability

The code used in this study is deposited at GitHub found at https://github.com/wesleypclawson/CEMA_Model.

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

## Acknowledgements

We thank Douglas Blackiston and other members of the Levin lab for helpful discussions, and Julia Poirier and Susan Lewis for help with the manuscript. This work was supported by the Emerald Gate Charitable Trust (awarded to M.L.). Research reported in this publication was supported by the National Institutes of Health via their support of a NovaSeq facility via award number S10OD032203 (awarded to M.L.). The authors gratefully acknowledge funding from the Army Research Office/DARPA under Cooperative Agreement Number W911NF-19-2-0027 (awarded to M.L.). The views and conclusions contained in this document are those of the authors and should not be interpreted as representing the official policies, either expressed or implied, of the Army Research Office/DARPA or the U.S. Government. The U.S. Government is authorized to reproduce and distribute reprints for Government purposes notwithstanding any copyright notation herein.

## Author contributions

A.T., M.M.S., and M.L. designed the experiments. A.T. conducted most of the experiments. M.M.S., W.C., S.B., and P.V.P. performed some experiments. A.P. and P.M. assisted A.T. with experiments and blind scoring. M.Y., R.M.F., and F.K. designed and 3D-printed constructs for experiments.

## Competing interests

M.L. is a scientific co-founder of and has an interest in, Morphoceuticals Inc. and Astonishing Labs, companies that operate in the regenerative medicine space—a field that could someday be impacted by the effect described in this manuscript. The remaining authors declare no competing interests.
