## [Peer Review File · Nature Communications]

Embryos assist morphogenesis of others through calcium and ATP signaling mechanisms in collective teratogen resistanceREVIEWER COMMENTS

Reviewer #1 (Remarks to the Author):

Review of manuscript NCOMMS-23-27594

Title: Embryos Assist Each Other's Morphogenesis: calcium and ATP signaling mechanisms in collective resistance to teratogens

Key results: The authors describe a novel effect where embryos respond to molecular and biological perturbations by releasing ATP. If an embryo is in close proximity to the ATP (as it would be in larger groups) it activates P2 receptors and through an unknown mechanism, but that involves intercellular calcium release, prevents abnormal development. The authors show several different types of cleverly designed experiments as well as computational modeling to support their model which they term Cross-Embryo Morphogenetic Assistance or CEMA. This work is unique and will have impact on a number of fields. Not only does it add a new layer to what we know about developmental mechanisms, but it will transform how experiments are designed in developmental biology and toxicology fields. It can also have important translation to understanding similar mechanisms in immune function.

Significance: This is a very transformative and novel study and I believe it should be published in this journal.

Validity: The authors provide several lines of evidence to support the validity of their model.

Data and methodology: The approach is comprehensive and of sufficient quality. Appropriate numbers, replicates and controls are included for all data. However, there could be improvements to the description of the methods and presentation of the data (see sections below for specific comments).

Analytical approach: The analytical approach and statistics used is appropriate. Although I am not an expert in statistics, I do feel competent to assess the approaches. On the other hand, I do NOT feel competent to assess the computational modeling.

A few clarifications in the approach and analyses would be useful:

1. In the analysis of thioridazine concentration in the media "there was no significant difference" however I don't see any statistical test or graphs in figure 1B. Was this measured in multiple experiments? If so, this information should be added. If not take out "significant" from the text and comment on the degree of variation ranging from 43 to 50 in the concentrations.
2. In the analysis of the craniofacial phenotype, the percentage of embryos with a square head is shown (fig 1E). Please provide a description in the methods of how an embryo was judged to have a square head. Is this a qualitative or quantitative assessment? Were there intermediates to the embryos shown?

In the results (or supplement) show more than one representative embryo.

4. In the methods there is a section called morphometrics where there is a description of head width measured between two points “(generally near the middle of the eyes)”. However, this is not what is depicted in fig1G. Fix the discrepancy and make the methods description more precise. Also, in this section there is a description for the angle of the head. I might have missed this but can’t find any data showing head angle. Remove this from the methods unless I am mistaken. If it is in the data an image showing landmarks should be included.

Suggested improvements: I do not think additional experiments are required. The data shown is comprehensive and sufficient to support this new finding and the proposed model. However, the depth of analysis, presentation and discussion should be improved which would lead to a stronger manuscript. My suggestions are as follows:

1. I request some additional information concerning the identity and function of the genes identified in the RNA-seq experiment. I would like to see a supplemental excel file that contains all genes identified as significant showing the padj value, the log fold change and the spelled-out name of the gene. All genes with a LOC... identifier should be looked up in NCBI and the gene name provided where possible. Additionally, for each *Xenopus* gene the corresponding human protein should be identified, and its function described. This can be found at genecards.org for example. From this you may be able to identify those genes associated with ATP and calcium (for example I looked at CRISP1 and there is an association with calcium channel function). The raw data should also be uploaded to a public data repository and this information should be included in the methods section.

2.. The authors do a great job demonstrating that CEMA occurs under a number of chemical/genetic perturbations. However, all affected molecules shown are associated with cellular communications (eg cAMP, transmitters, ion channels). I wonder if the same effects would occur in embryos that had other classes of molecules perturbed such as patterning transcription factors. In other words, do all types of developmental perturbations cause embryos to generate ATP to signal to their neighbors? This could be something to add to limitations and future work.

3. I think having both the teratogen exposures and the injury assay provides further evidence that inter-embryo communication is a generalized mechanism used by embryos. The injury assay is useful because you can control the timing of the insult. I wonder if a prolonged teratogen would be similar in its activation of calcium or would there be a slower effect that builds up over time. Please spell out the mechanistic connections of the two assays and the similarities and differences in the possible outcomes in the discussion.

4. The experiments using inhibitors of P2 and calcium effectively demonstrate the mechanism of CEMA but for someone who does not know much about P2 receptors it would be beneficial to provide more background. Therefore, please provide more information about P2 receptors in the discussion. What do they do in embryos and adults? For example, add some background about their roles in craniofacial development in zebrafish and their roles in immune function. Also add information about when and where P2 receptors are expressed in *Xenopus*. Xenbase can be used to assess temporal expression

(mRNA and protein) of each receptor subtype. In addition, it would also be beneficial to briefly discuss localization of P2 receptors reported in PMID: 31402854 (see below for full reference). Does enrichment of P2 receptors in craniofacial or brain tissues explain why these structures are influenced by CEMA?

5. Along the same lines as my train of thoughts in point 4.... In the experiments showing the differences in embryonic phenotypes in different group sizes, I was wondering if all phenotypes were reduced in the embryos in the larger groups. You show pictures of only heads but are there differences in overall size/length or other structures (heart, kidney, tail etc). If you have images of the full body please put these into the supplemental data. Please comment on effects in other regions of the embryo in the results section.

6. In the discussion it is implied that embryos communicate with each other and assist in morphogenesis. For example "Simple concentration of ATP or any other molecule does not have the bandwidth to specify the information needed for complex morphogenesis as observed here." I was wondering if there are alternatives to this idea? For example, could the CEMA effect be a response elicited by embryo to protect cells from the teratogen (or genetic manipulation, or injury/death in neighbors). So rather than assist with morphogenesis it prevents the disruption of morphogenesis. Maybe the increase in calcium in the embryo can activate endocytosis/lysosomal degradation pathways and reduce the concentration of the abnormal molecule in all or specific cells. Please present alternatives such as this briefly in the discussion.

Clarity and context: On the whole this manuscript is clearly written and informative. However, I think the figures could be clearer and have a more professional appearance. Therefore, I recommend some improvements to the figures to provide more clarity to readers.

General improvements:

1. In all figures I would like to see more labels to make the figures easier to read without searching through results and figure legends. Labeling such as:
 - a) Titles on all graphs.
 - b) Labels on pictures of embryos to differentiate control verses experimental (and what they were treated/injected with).
 - c) labeling some structures on the images of embryos to help orient the reader. Structures such as the eyes, brain would be useful. Orientation (anterior vs posterior) and view (dorsal or ventral) should be indicated in the legend.
 - d) show more than one representative embryo either in all the figures or in the supplemental data.

Specific problems with each figure include:

2. In figure 1 you should have an H on the last panel to be consistent with the rest of the graphs.

3. In figure 2, change the orientation of the embryos to be consistent with figure 1. The figures of

embryos should be lightened in panel B -they are too dark relative to panels A and C. The spacing between the pictures of the embryos should be consistent in all the panels, guidelines could be used. The scale bar in panel C needs to be moved. The red and blue arrowheads are not very informative and since they are so large they obscure some of the image. Instead, I would suggest labelling the structures in the wildtype and then use a smaller arrowhead to point out the difference in that structure in the experimental embryos. Clearly explain what the difference in the brains or eyes are in either the figure legend or results. What exactly is different in the brain structure, eyes etc?

4. Provide images of embryos for figures 3 and 4 and combine these figures into one figure similar to figure 2. One simple graph as a stand-alone figure doesn't seem substantial enough for this journal.

5. In figure 6, check that the font sizes big enough in panels C and D?

6. In figure 7, (this is only cosmetic) but the graph sizes seem rather large compared to other figures. I suggest that you try to make them consistent throughout the paper. Add some labels to images in panel A and describe the difference in the angle and what it shows (point out the windows for example). Also add some representative images of embryos as in fig 1 and 2 if you have them.

7. In Figure 8, add representative images of embryos to this panel as in figures 1 and 2.

8. In Figure 9, I am a little confused about the 1st image in panel A. There are two rows of wells. Is the second row of bigger wells used?

9. In the methods, in the microinjection section (page 38) clarify 2/2 and 4/4. I am guessing that 2/2 means 2 cells were injected at the 2-cell stage. There is also a "(Check)" in this section, remove this. Please also provide more specifics about the synthesis of mRNA (eg where is the plasmid from, the restriction enzyme and the polymerase utilized) or provide a reference where this is described elsewhere.

References: I recommend a few additional references be added that would accompany the addition of more information. The following should minimally be added.

Front Cell Neurosci. 2019 Jul 26;13:340. doi: 10.3389/fncel.2019.00340. eCollection 2019. Comparative Embryonic Spatio-Temporal Expression Profile Map of the Xenopus P2X Receptor Family. Camille Blanchard 1 2, Eric Boué-Grabot 1 2, Karine Massé 1 2
PMID: 31402854

Purinergic Signal. 2009 Sep; 5(3): 395–407.

Ectodermal P2X receptor function plays a pivotal role in craniofacial development of the zebrafish Sarah Kucenas,1,2 Jane A. Cox,1 Florentina Soto,1,3 Angela LaMora,1 and Mark M. Voigtcorresponding author1

PMID: 19529983

Your expertise: I do not have expertise in computational modeling and therefore I do not make any

comments on this section (in both the results and discussion).

Reviewer #2 (Remarks to the Author):

In the manuscript titled “Embryos assist each other’s morphogenesis: calcium and ATP signaling mechanisms in collective resistance to teratogens” authors investigated the tantalizing possibility that embryo-embryo communication may assist the population in dealing with stress-induced injuries.

Authors indeed report a higher survival and decreased defect rate in larger cohorts, with an optimal size of 300 embryos. To conceptualize their findings further they present an agent-based model in which agents performance is assessed by their ability to arrive to a sequence of either all 1s or all 0s representing the “majority” result from a starting random sequence of 1s and 0s.

Overall I find the study well-motivated and most of the results are rigorous and convincing. However, I also have a few major questions and points of concern that I believe should be addressed for the paper to be accepted for publication in Nature Communication.

Major points:

In Calcium depletion experiments presented in Figure 8, are used to argue for the importance of Ca in mediating signaling. It seems to me that important controls of Thapsigargin(Thg) alone or PPADS alone are missing. These are likely very toxic on their own and presented results of decreased survival may stem from intra-embryo toxicity of Thg/PPADS.

While I find the experiments on signal transmission from a mechanically injured embryo to an uninjured neighbor exciting, the current presentation of the results does not appear convincing.

Authors state “In many cases, we observed that the calcium wave in the second embryo starts at a point adjacent to the injured embryo, however, the wave is not completely consistent in starting at this closest point”. This statement calls for quantification and statistical significance and I am surprised authors did not present the quantification of this observation (e.g. the distance from the center of the injured embryo to the “point of entry” in uninjured, this can then be compared to the distance to the random point in uninjured neighbor and estimate p-value for difference between the two distance.)

It is unclear why authors choose to present the results on the wave speed . What question or hypothesis does this result address? Is it to support that the wave is propagating from injured to uninjured embryo? Related to the above, it would be very informative to see how the average intensity of Ca activity changes with time in Pairs of injured-receiver embryos. It seems that the time profiles of mean intensity should be a more appropriate measure (e.g. and it is therefore not clear why authors choose to report the max intensity in Figure 9E-G. It is very likely becomes apparent why max intensity is appropriate when time-series of the intensities are presented as well.

In addition, to make a convincing case of signal propagation, it is important to compare the “injured-uninjured receiver” pairs with the control where pairs of both embryos are uninjured.

I am also concerned that the significant difference Figure 9F may come from larger number of samples (N=10) in control group vs. N=5 in suramin group. Even with random numbers drawn, larger number of samples means higher chance some of them will be coming from the tails of distribution. In case of data in Figure 9F this seems to be a distribution that's bounded by a value close to 1 and therefore larger number of samples would result in larger means, as the distribution is not bounded at the upper end. While I find modeling results very relevant, I am missing a few important points. It is unclear how a signal, such as e.g. Ca or ATP can carry the information that would allow the embryo to correct the deleterious effects on the cellular or sub-cellular level. Is the hypothesis that this is more of a "warning" signal, allowing embryos to activate protective mechanism on par with insect injured plants sending pheromones that then allows neighboring plants to mount a defense in advance. This point actually applies to the results of the entire manuscript, i.e. it would be very helpful with a discussion on how the inter-embryo Ca-mediated signaling could be used through the known mechanisms guiding the development.

While the model is minimal and rather well described, it is not clear how sensitive is the model outcomes to the specific choice of the parameters? E.g. can the same results be obtained if one assumes only nearest neighbor interactions?

Is the threshold of 0.5 on sending the signal critical and what happens if it is lowered or set to 0?

Minor points:

Authors mention that the "Media and dish size was scaled proportionally between large and small group...". without specifying (neither main text or methods as far as I could see) how exactly the scaling was done. An explanation and a formula relating group size with volume of the media and size of the dish would help understand why this was needed. While it certainly helps to see the thioridazine concentration remained constant across the differently-sized groups, it remains unclear if scaling was necessary

Reviewer #1 (Remarks to the Author):

Significance: This is a very **transformative and novel study and I believe it should be published in this journal.**

Validity: The authors provide several lines of evidence to support the validity of their model.

Data and methodology: The approach is comprehensive and of sufficient quality. Appropriate numbers, replicates and controls are included for all data. However, there could be improvements to the description of the methods and presentation of the data (see sections below for specific comments).

We thank the reviewer for their highly positive assessment and very helpful suggestions. We have made the suggested improvements, as itemized below.

1. In the analysis of thioridazine concentration in the media “there was no significant difference” however I don’t see any statistical test or graphs in figure 1B. Was this measured in multiple experiments? If so, this information should be added. If not take out “significant” from the text and comment on the degree of variation ranging from 43 to 50 in the concentrations.

Done. We have removed the word significant and addressed a potential source of variation.

2. In the analysis of the craniofacial phenotype, the percentage of embryos with a square head is shown (fig 1E). Please provide a description in the methods of how an embryo was judged to have a square head. Is this a qualitative or quantitative assessment? Were there intermediates to the embryos shown? In the results (or supplement) show more than one representative embryo.

Under the morphometric section of the methods, we now describe the measurements we used to define how we score for square heads. Additionally, figure 1G now shows measurements taken to measure squareness. There is some variation in squareness and we now provide additional photos in the Supplement.

4. In the methods there is a section called morphometrics where there is a description of head width measured between two points “(generally near the middle of the eyes)”. However, this is not what is depicted in fig1G. Fix the discrepancy and make the methods description more precise. Also, in this section there is a description for the angle of the head. I might have missed this but can’t find any data showing head angle. Remove this from the methods unless I am mistaken. If it is in the data an image showing lands marks should be included.

We apologize for the discrepancy, and have reworded the methods section to make it more clear; we have also removed the section on head angle because it was confusing and not necessary to the claims.

Suggested improvements: I do not think additional experiments are required. The data shown is comprehensive and sufficient to support this new finding and the proposed model. However, the depth of analysis, presentation and discussion should be improved which would lead to a stronger manuscript. My suggestions are as follows:

1. I request some additional information concerning the identity and function of the genes identified in the RNA-seq experiment. I would like to see a supplemental excel file that contains all genes identified as significant showing the padj value, the log fold change and the spelled-out name of the gene. All genes with a LOC... identifier should be looked up in NCBI and the gene name provided where possible. Additionally, for each *Xenopus* gene the corresponding human protein should be identified, and its function described. This can be found at genecards.org for example. From this you may be able to identify those genes associated with ATP and calcium (for example I looked at CRISP1 and there is an association with calcium channel function). The raw data should also be uploaded to a public data repository and this information should be included in the methods section.

Done. All the significantly differentially expressed genes are now listed in a supplemental excel file (Supplemental File 1) that includes the padj, log fold change, spelled out name of the gene, and the corresponding human protein and function. In that file, we also include the gene name for each LOC, where possible. For each of the genes, we also looked more deeply for any potential associations with ATP and calcium. Thank you for this insight into the CRISP1 function. These functions have been added to the table as well. Finally, we have uploaded the RNA-seq raw data to GEO as a Super Series that includes both the Stage 25 and 35 datasets.

2. The authors do a great job demonstrating that CEMA occurs under a number of chemical/genetic perturbations. However, all affected molecules shown are associated with cellular communications (eg cAMP, transmitters, ion channels). I wonder if the same effects would occur in embryos that had other classes of molecules perturbed such as patterning transcription factors. In other words, do all types of developmental perturbations cause embryos to generate ATP to signal to their neighbors? This could be something to add to limitations and future work.

This is a great point and we have now mentioned it in the Discussion.

3. I think having both the teratogen exposures and the injury assay provides further evidence that inter-embryo communication is a generalized mechanism used by embryos. The injury assay is useful because you can control the timing of the insult. I wonder if a prolonged teratogen would be similar in its activation of calcium or would there be a slower effect that builds up over time. Please spell out the mechanistic connections of the two assays and the similarities and differences in the possible outcomes in the discussion.

The two assays are similar in that they are both inducing injury/stress onto the embryo; however, one is a mechanical injury while the other is chemical. Timescales are also different where the injury is 30 minutes while the teratogen is 15 hours. Furthermore, the scale in which these perturbations occur are different. The mechanical injury is an example of a more discrete event with a known timepoint, while the teratogen is a more persistent insult stretched out over time. We now mention this in the Discussion and point out the relative advantages of each and why we used them both.

4. The experiments using inhibitors of P2 and calcium effectively demonstrate the mechanism of CEMA but for someone who does not know much about P2 receptors it would be beneficial to provide more background. Therefore, please provide more information about P2 receptors in the discussion. What do they do in embryos and adults? For example, add some background about their roles in craniofacial development in zebrafish and their roles in immune function. Also add information about when and where P2 receptors are expressed in *Xenopus*. Xenbase can be used to assess temporal expression (mRNA and protein) of each receptor subtype. In addition, it would also be beneficial to briefly discuss localization of P2 receptors reported in PMID: 31402854 (see below for full reference). Does enrichment of P2 receptors in craniofacial or brain tissues explain why these structures are influenced by CEMA?

Front Cell Neurosci. 2019 Jul 26;13:340. doi: 10.3389/fncel.2019.00340. eCollection 2019. Comparative Embryonic Spatio-Temporal Expression Profile Map of the *Xenopus* P2X Receptor Family. Camille Blanchard 1 2, Eric Boué-Grabot 1 2, Karine Massé 1 2 PMID: 31402854

Purinergic Signal. 2009 Sep; 5(3): 395–407. Ectodermal P2X receptor function plays a pivotal role in craniofacial development of the zebrafish
Sarah Kucenas,1,2 Jane A. Cox,1 Florentina Soto,1,3 Angela LaMora,1 and Mark M. Voigt corresponding author1 PMID: 19529983

Knockdown of P2X3.1 resulted in smaller heads with malformed lower jaw and branchial arch regions. The smaller heads could be a result of defective pharyngeal skeleton development. The loss of p2rx3.1 signaling results in atypical neural crest behaviors in the branchial arches as well. We now mention the above information in the Discussion.

P2 receptors have been detected in the central and peripheral nervous system of *Xenopus laevis* with overlapping expression. While each subunit has a distinct expression profile, there are several of them that are expressed in the head region. At stages 27 and 33/34, the p2rx1.L subunit is localized to a specific region of the hind brain as well as in the head region, in the branchial arches and in the cement gland. Expression of the p2rx1.S starts at the cleavage stage but localizes to the eye field during neurulation till stage 41. The expression of p2rx2.L is mostly found in developing mesoderm derivatives and nervous system. Expression can be seen in the head region, the brain, and sensory organs like the eyes at stage 41. Finally, a third member of the p2x receptor family is weakly found at the somites (stage 33/34), p2rx6.L. At stage 41, this receptor expression is strong in the head and dorsal regions. Temporally, the start of expression may be unique, but the level of expression for all of them increases during development and reaches a peak at stage 45. We now mention all this in the Discussion, in the context of our phenotypes.

5. Along the same lines as my train of thoughts in point 4.... In the experiments showing the differences in embryonic phenotypes in different group sizes, I was wondering if all phenotypes were reduced in the embryos in the larger groups. You show pictures of only heads but are there differences in overall size/length or other structures (heart, kidney, tail etc). If you have images of the full body please put these into the supplemental data. Please comment on effects in other regions of the embryo in the results section.

While there are other types of defects that form such as edemas and kinked bodies, we chose to focus on craniofacial defects in this paper. We also now have mentioned the other regions in the Results and included them in a new Supplement.

6. In the discussion it is implied that embryos communicate with each other and assist in morphogenesis. For example "Simple concentration of ATP or any other molecule does not have the bandwidth to specify the information needed for complex morphogenesis as observed here." I was wondering if there are alternatives to this idea? For example, could the CEMA effect be a response elicited by embryo to protect cells from the teratogen (or genetic manipulation, or injury/death in neighbors). So rather than assist with morphogenesis it prevents the disruption of morphogenesis. Maybe the increase in calcium in the embryo can activate endocytosis/lysosomal degradation pathways and reduce the concentration of the abnormal molecule in all or specific cells. Please present alternatives such as this briefly in the discussion.

Done; we have now mentioned this in the Discussion.

Clarity and context: On the whole this manuscript is clearly written and informative. However, I think the figures could be clearer and have a more professional appearance. Therefore, I recommend some improvements to the figures to provide more clarity to readers. General improvements:

1. In all figures I would like to see more labels to make the figures easier to read without searching through results and figure legends. Labeling such as:

- a) Titles on all graphs
- b) Labels on pictures of embryos to differentiate control verses experimental (and what they were treated/injected with)
- c) labeling some structures on the images of embryos to help orient the reader. Structures such as the eyes, brain would be useful. Orientation (anterior vs posterior) and view (dorsal or ventral) should be indicated in the legend
- d) show more than one representative embryo either in all the figures or in the supplemental data.

Done. We have now made the suggested improvements.

Specific problems with each figure include:

2. In figure 1 you should have an H on the last panel to be consistent with the rest of the graphs.

Done.

3. In figure 2, change the orientation of the embryos to be consistent with figure 1. The figures of embryos should be lightened in panel B -they are too dark relative to panels A and C. The spacing between the pictures of the embryos should be consistent in all the panels, guidelines could be used. The scale bar in panel C needs to be moved. The red and blue arrowheads are not very informative and since they are so large they obscure some of the image. Instead, I would suggest labelling the structures in the wildtype and then use a smaller arrowhead to point out the difference in that structure in the experimental embryos. Clearly explain what the difference in the brains or eyes are in either the figure legend or results. What exactly is different in the brain structure, eyes etc?

Done.

4. Provide images of embryos for figures 3 and 4 and combine these figures into one figure similar to figure 2. One simple graph as a stand-alone figure doesn't seem substantial enough for this journal.

Done.

5. In figure 6, check that the font sizes big enough in panels C and D?

Done.

6. In figure 7, (this is only cosmetic) but the graph sizes seem rather large compared to other figures. I suggest that you try to make them consistent throughout the paper. Add some labels to images in panel A and describe the difference in the angle and what it shows (point out the windows for example). Also add some representative images of embryos as in fig 1 and 2 if you have them.

Done.

7. In Figure 8, add representative images of embryos to this panel as in figures 1 and 2.

Done.

8. In Figure 9, I am a little confused about the 1st image in panel A. There are two rows of wells. Is the second row of bigger wells used?

The second row of bigger wells was not used in this paper. To minimize confusion, we have edited this figure to show only the left hand column of wells.

9. In the methods, in the microinjection section (page 38) clarify 2/2 and 4/4. I am guessing that 2/2 means 2 cells were injected at the 2-cell stage. There is also a "(Check)" in this section, remove this. Please also provide more specifics about the synthesis of mRNA (eg where is the plasmid from, the restriction enzyme and the polymerase utilized) or provide a reference where this is described elsewhere.

Done. We have clarified in the Methods that 2 of 2 cells and 4 of 4 cells were injected as well as removed the (Check) note. We have also provided references where the mRNA synthesis and injection protocol are described.

References: I recommend a few additional references be added that would accompany the addition of more information. The following should minimally be added.

Front Cell Neurosci. 2019 Jul 26;13:340. doi: 10.3389/fncel.2019.00340. eCollection 2019. Comparative Embryonic Spatio-Temporal Expression Profile Map of the Xenopus P2X Receptor Family. Camille Blanchard 1 2, Eric Boué-Grabot 1 2, Karine Massé 1 2 PMID: 31402854

Purinergic Signal. 2009 Sep; 5(3): 395–407. Ectodermal P2X receptor function plays a pivotal role in craniofacial development of the zebrafish

Sarah Kucenas,1,2 Jane A. Cox,1 Florentina Soto,1,3 Angela LaMora,1 and Mark M. Voigt corresponding author1 PMID: 19529983

We thank the reviewer for these references and have now incorporated them.

Reviewer #2 (Remarks to the Author):

In the manuscript titled “Embryos assist each other’s morphogenesis: calcium and ATP signaling mechanisms in collective resistance to teratogens” authors investigated the tantalizing possibility that embryo-embryo communication may assist the population in dealing with stress-induced injuries. Authors indeed report a higher survival and decreased defect rate in larger cohorts, with an optimal size of 300 embryos. To conceptualize their findings further they present an agent-based model in which agents performance is assessed by their ability to arrive to a sequence of either all 1s or all 0s representing the “majority” result from a starting random sequence of 1s and 0s. **Overall I find the study well-motivated and most of the results are rigorous and convincing.**

We thank the reviewer for this highly positive assessment.

However, I also have a few major questions and points of concern that I believe should be addressed for the paper to be accepted for publication in Nature Communication.

We are grateful for the thoughtful comments and have addressed them as follows:

In Calcium depletion experiments presented in Figure 8, are used to argue for the importance of Ca in mediating signaling. It seems to me that important controls of Thapsigargin(Thg) alone or PPADS alone are missing. These are likely very toxic on their own and presented results of decreased survival may stem from intra-embryo toxicity of Thg/PPADS.

This is a good point; the necessary controls have now been added to refute this possibility. Thapsigargin was lethal on its own and has been removed.

While I find the experiments on signal transmission from a mechanically injured embryo to an uninjured neighbor exciting, the current presentation of the results does not appear convincing. Authors state “In many cases, we observed that the calcium wave in the second embryo starts at a point adjacent to the injured embryo, however, the wave is not completely consistent in starting at this closest point”. This statement calls for quantification and statistical significance and I am surprised authors did not present the quantification of this observation (e.g. the distance from the center of the injured embryo to the “point of entry” in uninjured, this can then be compared to the distance to the random point in uninjured neighbor and estimate p-value for difference between the two distance.)

We have now removed any claims about the position of the start point of the response wave. It has become clear that this is a complex and fascinating issue that needs its own paper. It is not necessary to support any of our main claims here, and as the reviewer points out, requires a lot more work to nail down the real relationship. Therefore, we will avoid making any geometric claims here about the wave and address it in full detail in a subsequent paper.

It is unclear why authors choose to present the results on the wave speed. What question or hypothesis does this result address? Is it to support that the wave is propagating from injured to uninjured embryo?

We do not make any claims based on the wave speed; however, this being a new phenomenon in this context, we thought it prudent to provide as much descriptive quantitative characterization as possible. This may be helpful for comparison to other types of biological signaling, including conduction velocity in the heart (2-5 meters/second), cranial blood flow (24-100 cm/s), and neuronal action potentials (0.5-100 m/s). In our lab, we have also observed long-range signaling between injured and uninjured limbs within 5 seconds of amputation (these are now cited in the manuscript). We have incorporated this information into the Discussion to put the injury wave speed into context. We can remove it, if the editor feels that it interferes, but we believe readers will wonder how fast the wave is (for example, to possibly utilize it in bioengineering contexts, to compare its speed to their own favorite signaling modality, etc.), so our preference is to leave it as-is, but we are certainly flexible on this.

Related to the above, it would be very informative to see how the average intensity of Ca activity changes with time in Pairs of injured-receiver embryos. It seems that the time profiles of mean intensity should be a more appropriate measure (e.g. and it is therefore not clear why authors choose to report the max intensity in Figure 9E-G. It is very likely becomes apparent why max intensity is appropriate when time-series of the intensities are presented as well.

We averaged the signal over each embryo and reported the maximum mean signal observed across each embryo over time. This approach allowed for a single metric by which we can compare the extent of signal propagation achieved under both normal and treated conditions. However, we also appreciate that there is a critical time component to signal propagation and therefore showed this aspect in the form of kymographs (Figure 8D). To provide further clarification, we now include the average intensity of calcium changes over time as well (Supplemental Figure 4).

In addition, to make a convincing case of signal propagation, it is important to compare the “injured-uninjured receiver” pairs with the control where pairs of both embryos are uninjured

All of the data presented is normalized to baseline, spontaneous data for each pair of embryos. This approach allows us to have embryo-specific controls for normalization of the intensity levels, thereby correcting for any natural variation in GCamp6s expression. In addition, in the Supplementary figures we now provide intensity vs. time plots (Supplemental Figure 4), which include 10 minutes of calcium activity that was captured for each embryo prior to injury. During this period, we don't observe any significant calcium transients that allow us to clearly conclude that uninjured embryos communicate. However, we can't rule out that there is some level of signaling that we can't capture.

I am also concerned that the significant difference Figure 9F may come from larger number of samples (N=10) in control group vs. N=5 in suramin group. Even with random numbers drawn, larger number of samples means higher chance some of them will be coming from the tails of distribution. In case of data in Figure 9F this seems to be a distribution that's bounded by a value close to 1 and therefore larger number of samples would result in larger means, as the distribution is not bounded at the upper end.

Thank you for bringing this to our attention. We have a higher number of controls in our database since we ran experiments with only controls to characterize the injury wave and controls are also included in treatment experiments as well. To ensure the reliability of our comparison, we used Welch's t-test, which is a more robust test for

comparing groups with unequal sample sizes and variance than the Student's T-test (Derrick et al. 2016). To directly test the impact of group size, we also manually sub-sampled controls in groups of N=5 and found that the smaller group size did not substantially impact the trend we observed ($p=0.02-0.16$ were measured across sub-samplings).

While I find modeling results very relevant, I am missing a few important points. It is unclear how a signal, such as e.g. Ca or ATP can carry the information that would allow the embryo to correct the deleterious effects on the cellular or sub-cellular level. Is the hypothesis that this is more of a "warning" signal, allowing embryos to activate protective mechanism on par with insect injured plants sending pheromones that then allows neighboring plants to mount a defense in advance. This point actually applies to the results of the entire manuscript, i.e. it would be very helpful with a discussion on how the inter-embryo Ca-mediated signaling could be used through the known mechanisms guiding the development.

This is a great question, and we now devote more space to it in the Discussion, as well as provide new data from additional experiments (Figure 3D). There are 2 possibilities, as the reviewer notes: the message could be a generic warning signal, or bear specific instructive morphogenetic information relevant to fixing a specific defect. To distinguish the two, we performed new experiments with mixed groups of embryos exposed to *different* teratogens. We found that they do not cross-protect, suggesting the signal is specific and instructive.

While the model is minimal and rather well described, it is not clear how sensitive is the model outcomes to the specific choice of the parameters? E.g. can the same results be obtained if one assumes only nearest neighbor interactions? Is the threshold of 0.5 on sending the signal critical and what happens if it is lowered or set to 0?

This is an interesting point, and we thank the reviewer for bringing it up, as our investigation of it has revealed some new insights which we've added to the manuscript. To address the first point, we ran the model in two different configurations; first, with the neighbor's neighbors' interactions (as submitted) and second, with only nearest neighbors and found that nearest neighbors is not sufficient to resist the effects of noise. Even when the noise was significantly reduced ($p=0.6$), there was not group-wide survival, but instead small pockets of survival between spaces of 'dead cells'. This further supports the CEMA effect, as small cohorts don't have neighbor's neighbors', and therefore are in similar conditions. To address the second point, we re-ran the original simulations with the threshold set at 0.75 and 0.25 and observed that both influenced the simulations,

with the higher threshold causing group 'death' far too quickly and the low threshold causing the recovery mechanism to active too quickly. This may be an avenue for further studies, as signaling thresholds may be a target of specific chemical interventions. Both results have been placed into Supplementary Figure 4.

Minor points:

Authors mention that the "Media and dish size was scaled proportionally between large and small group...". without specifying (neither main text or methods as far as I could see) how exactly the scaling was done. An explanation and a formula relating group size with volume of the media and size of the dish would be help understand why this was needed. While it certainly helps to see the thioridazine concentration remained constant across the differently-sized groups, it remains unclear if scaling was necessary.

Done. We now include text on exactly how it was done (in the Methods). The reason we scaled everything was to remove all other potential differences that could, formally, be proposed to be the source of difference. This is a rather striking phenomenon and we expected initially skeptical readers to try to find some other aspect responsible for the differences we report. We wanted to confirm, in an airtight fashion (leaving no other known factors, no matter how unlikely), that it was just the cohort size.

REVIEWERS' COMMENTS

Reviewer #1 (Remarks to the Author):

I believe the authors have sufficiently addressed all of my previous reviewer comments and therefore recommend this for publication without further changes.

Reviewer #2 (Remarks to the Author):

The revisions have addressed all my concerns and I believe that the manuscript, and its fascinating results, are ready for publication.